# Hierarchy of many-body invariants and quantized magnetization in anomalous Floquet insulators

Frederik Nathan[1], Dmitry A. Abanin[2], Netanel H. Lindner[3], Erez Berg[4,5], and Mark S. Rudner[1]

[1]*Center for Quantum Devices, Niels Bohr Institute,*
*University of Copenhagen, 2100 Copenhagen, Denmark*
[2]*Department of Theoretical Physics, University of Geneva, 1211 Geneva, Switzerland*
[3]*Physics Department, Technion, 320003 Haifa, Israel*
[4]*Department of Physics, University of Chicago, Chicago, IL 60637, USA*
[5]*Department of Condensed Matter Physics, The Weizmann Institute of Science, Rehovot, 76100, Israel*
(Dated: February 24, 2021)

We uncover a new family of few-body topological phases in periodically driven fermionic systems in two dimensions. These phases, which we term correlation-induced anomalous Floquet insulators (CIAFIs), are characterized by quantized contributions to the bulk magnetization from multi-particle correlations, and are classified by a family of integer-valued topological invariants. The CIAFI phases do not require many-body localization, but arise in the generic situation of $k$-particle localization, where the system is localized (due to disorder) for any finite number of particles up to a maximum number, $k$. We moreover show that, when fully many-body localized, periodically driven systems of interacting fermions in two dimensions are characterized by a quantized magnetization in the bulk, thus confirming the quantization of magnetization of the anomalous Floquet insulator. We demonstrate our results with numerical simulations.

In recent years, periodic driving has been studied as a means for realizing topological phases of matter [1–14]. An important result of this work has been the discovery of a wide range of intrinsically nonequilibrium topological phases with no equilibrium counterparts [14–37]. These "anomalous" phases are characterized by robust properties of their micromotion (i.e., the dynamics that takes place within a driving period), such as frequency-locked oscillations in Floquet time crystals [24–28], or quantized orbital magnetization density in the two-dimensional anomalous Floquet-Anderson insulator (AFAI) [16, 30, 31].

Disorder plays a crucial role for stabilizing Floquet phases in closed systems. In particular, in the presence of interactions, disorder-induced many-body localization (MBL) provides a mechanism for the system to avoid uncontrollably absorbing energy from the driving field, and thereby to retain nontrivial properties at long times [38–40]. Importantly, the requirement of many-body localization does not preclude the system from exhibiting a variety of types of symmetry-breaking and topological order [25, 26, 37].

In this paper we characterize the topological properties of time-evolution in two-dimensional periodically driven systems of fermions which exhibit either full many-body localization, or a weaker form of "$k$-particle localization" that we define below [37–40] (see Fig. 1). Recent results suggest that this class of systems can support a nontrivial topological phase, known as the Anomalous Floquet Insulator [37] (AFI), which can be seen as the generalization of the AFAI to interacting systems (see Refs. 30 and 31). Despite being localized and insulating, the AFI features nontrivial circulating currents in the bulk, which in the noninteracting case (the AFAI) give rise to quantized orbital magnetization [30]. In a geometry with boundaries, the AFI supports thermalizing chiral edge states

coexisting with a localized bulk [31, 37]. The existence AFI as a stable many-body state of matter rests on the existence of MBL; even if MBL does hold out to infinite times, the phenomenology of the AFI is expected to persist for at least exponentially long times.

The motivation of our work is to determine the topological invariant(s) that characterize the AFI. Focusing on the topological characterization of the micromotion of particles in the bulk (i.e., the dynamics which take place within each driving period), we uncover two main results.

As our first result, we confirm that, like the AFAI, the AFI is characterized by a quantized magnetization density in regions of the bulk where all states are occupied, as schematically depicted in Fig. 1a. Specifically, the magnetization density is quantized as $\mu_1/T$ where $T$ denotes the driving period, and $\mu_1$ is an integer characterizing the topological phase. This quantization is protected by many-body localization, and $\mu_1$ cannot change under any deformation of the system that preserves MBL.

As the second major finding of our work, we uncover a rich new structure of topological invariants that emerges in the interacting case: while periodically driven systems of noninteracting fermions in two dimensions (such as the AFAI) may be characterized by a single invariant $\mu_1$, their interacting counterparts are characterized by a *family* of integer-valued topological invariants $\mu_1, \mu_2, \ldots$. The invariant $\mu_\ell$ encodes information about the contribution to the time-averaged magnetization from $\ell$-particle correlations. Hence, interactions allow for a richer topological structure in the system.

The topological protection of the invariant $\mu_\ell$ relies on a less restrictive notion of localization than the conventional notion of MBL. Specifically, $\mu_\ell$ is well-defined and topologically protected when all Floquet eigenstates with up to $k$ particles are localized for some $k \geq l$. We term this notion of localization "$k$-particle localization."

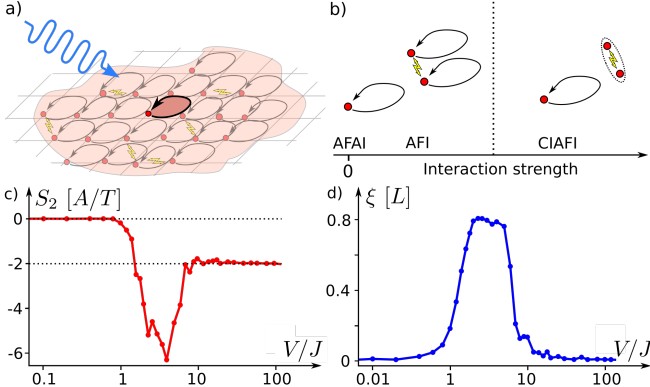

FIG. 1. (a) The anomalous Floquet insulator (AFI) is characterized by drive-induced circulating motion of particles in the bulk. Nontrivial topology is revealed in a quantized, nonzero magnetization density within regions where all states are filled, given by $\langle m \rangle = \frac{\mu_1}{T}$, where $\mu_1$ is a nonzero integer. (b) With sufficiently strong interactions, a new class of interaction-induced topological phases can emerge, which we term correlation-induced anomalous Floquet insulators (CIAFI's). CIAFI phases are characterized by a quantized, nonzero contribution to the magnetization from $\ell$-particle correlations. Such correlations can for example arise due to immobilization of many-particle bound states, as depicted in the figure. (c,d) Topological phase transition between the AFI and a CIAFI phase with $\mu_2 = 2$ obtained from numerical simulations of a driven Hubbard-like model (see Sec. IV for details). (c) Contribution to the time-averaged magnetization in the system due to two-particle correlations, $S_2$ (see Sec. I for definition and relationship with $\mu_2$), as a function of the interaction strength $V$. (d) The correlation length $\xi$ in the system diverges for interaction strength $V$ comparable to the hopping $J$, indicating a topological transition between AFI and CIAFI phases.

Many-body localization corresponds to $k$-particle localization in the limit where $k$ and the system size goes to infinity, while allowing the particle density to be finite in the thermodynamic limit. While the existence of MBL in more than one dimension is still a subject of debate [41], $k$-particle localization for finite $k$ is well established in any dimension [42]. It is likely that systems exhibiting $k$-particle localization, even if not fully MBL, may still display long-lived transient phenomena: delocalization in such systems must be induced by $k+1$-particle correlated processes, whose rates are expected to be exponentially suppressed in $k$ for sufficiently weak interactions.

Our results above show that $k$-particle localized Floquet systems of interacting fermions in 2D are characterized by $k$ independent topological invariants, $\mu_1, \ldots \mu_k$. When one or more of the higher-order invariants are nonzero, the system is in a new, strongly-correlated, intrinsically nonequilibrium phase that is topologically distinct from any noninteracting system, including the (noninteracting) AFAI. We term this class of phases Correlation-Induced AFIs (CIAFIs). Here we consider a broader notion of the term "phase" than for equilibrium systems; in the sense we consider here, a phase charac-

terizes the structure of the Hamiltonian of the *isolated* system, independently of the particular state of the system (and in particularly, independently of particle density and temperature).

We present a family of models which interpolate from the AFI phase to a CIAFI phase with a nonzero value of $\mu_2$, and demonstrate the existence of a nontrivial CIAFI phase in the model through numerical simulations [see Fig. 1(c)-(d)].

The arguments leading to the identification of the higher-order invariants $\mu_\ell$ can in principle also be applied to bosonic systems where the total number of bosons is conserved (e.g., as in systems of bosonic atoms in optical lattices). Hence AFI and CIAFI phases also exist for $k$-particle localized bosonic systems. However, for simplicity, in this paper, we consider fermionic systems only.

The rest of the paper is organized as follows. In Sec. I, we summarize the main results of this paper. In Sec. II we briefly review the structure of the Floquet operator in many-body and $k$-particle localized systems, and of the orbital magnetization operator. In Sec. III we use the time-averaged magnetization density operator to identify a set of topological invariants $\{\mu_\ell\}$ that characterize the AFI phase, and show that nonzero values of the invariants give rise to a quantized magnetization density in regions where all sites are occupied (Sec. III D). In Sec. IV we present a family of models that realize both the AFI and CIAFI phases, and support our conclusions with numerical simulations of these models. We conclude with a discussion in Sec. V.

## I. SUMMARY OF MAIN RESULTS

We begin by summarizing the main results of this paper. We consider a two-dimensional periodically driven systems of interacting fermions, which is $k$-particle (or many-body) localized due to disorder [43]. To characterize the topology of the system, we quantify the circulating motion of particles in the bulk. This circulating motion can be captured through the time-averaged magnetization density operator of each plaquette $p$ in the Heisenberg picture, $\bar{m}_p$. The magnetization density $\bar{m}_p$ measures the total time-averaged current that circulates around the plaquette; see Sec. II for a definition of this operator and a review of its properties. From its intrinsic properties, we show that the trace of $\bar{m}_p$ defines a family of topological invariants for the system. Specifically, the trace of $\bar{m}_p$ in the $\ell$-particle subspace, $\mathrm{Tr}_\ell \bar{m}_p$, for each $\ell = 1, \ldots k$, must take the same value for each plaquette in the system; this value cannot change under any smooth deformation of the parameters of the system that preserves $k$-particle localization. Hence $\mathrm{Tr}_\ell \bar{m}_p$ for each $\ell = 1, \ldots k$ constitutes a topological invariant of the system. The intrinsic invariants $\mu_1 \ldots \mu_k$ described in the introduction are constructed by forming system-size independent, integer-valued combinations of the (system size dependent) invariants $\mathrm{Tr}_1 \bar{m}_p, \ldots \mathrm{Tr}_k \bar{m}_p$; see Sec. III C

for further details.

To illustrate the physical meaning of the invariants $\{\mu_\ell\}$, consider first the case where the system holds a single fermion, initially located on site $i$ in the lattice (we assume, without loss of generality, that each site holds a single orbital). When all single-particle Floquet eigenstates are localized, the particle will remain confined near site $i$ at all times. However, the driving field may cause the particle to undergo circulating motion, as schematically depicted in the bottom left of Fig. 1(b). This circulating motion gives rise to a nonzero long-time-averaged (orbital) moment, $\bar{M}_i$. For both single- and many-particle systems (which we consider below), the total time-averaged magnetic moment can be computed as the integral of magnetization density over the entire lattice, $\sum_p \bar{m}_p a^2$. Ref. [31] showed that the sum of $\bar{M}_i$ over all single-particle states, $S_1 \equiv \sum_i \bar{M}_i$, is quantized as an integer times $A/T$, where $A$ denotes the area of the system; this integer defines $\mu_1$. As an implication, magnetization density is quantized in the bulk of the system in regions where all states are occupied.

We now consider the dynamics resulting from initializing the system in a two-particle state where sites $i$ and $j$ are occupied. We let $\bar{M}_{ij}$ denote the total long-time-averaged magnetization of the system resulting from this initialization. In the absence of interactions, one can verify that $\bar{M}_{ij} = \bar{M}_i + \bar{M}_j$. However, with interactions present, $\bar{M}_{ij}$ generically differs from $\bar{M}_i + \bar{M}_j$ when sites $i$ and $j$ are close to each other. The deviation can be measured by the "magnetization cumulant" $C_{ij} \equiv \bar{M}_{ij} - (\bar{M}_i + \bar{M}_j)$. In Sec. III below, we show that, when all 1- and 2-particle states are localized, the sum of $C_{ij}$ over all distinct two-particle configurations, $S_2 \equiv \sum_{i<j} C_{ij}$, must be *quantized*, as an integer $\mu_2$ times $A/T$. The number $\mu_2$ cannot change under any perturbation that preserves localization of states with 1 and 2 particles. Thus, $\mu_2$ is a topological invariant protected by 2-particle localization, and characterizes the contribution to the magnetization associated with 2-particle correlations. The higher-order invariants, $\mu_\ell$ for $\ell > 2$, are defined analogously to $\mu_2$ from higher-order "cumulants" of the magnetization (see Sec. III C for details), and $\mu_\ell$ is protected under any perturbation that preserves $\ell$-particle localization.

We term the class of phases characterized by nonzero values of the higher-order invariants (i.e., $\mu_\ell$ for $\ell > 1$) as correlation-induced anomalous Floquet insulators (CIAFIs). The AFI phase is the MBL extension of the noninteracting AFAI, where all higher-order invariants must be zero, and can thus only be characterized by a nonzero value of $\mu_1$. Hence the CIAFI phases are distinct from the AFI.

In Sec. IV we present a model that realizes a CIAFI phase with $\mu_2 = -2$. The model consists of spin-1/2 fermions on a bipartite square lattice with Hubbard-like on-site interactions and disorder, subject to the 5-step driving protocol of the canonical AFAI model [16, 30, 31] [see Fig. 3(a)].

As discussed in Sec. IV, and shown numerically in Fig. 1(c), the strength of the Hubbard-type interactions, $V$, controls the topological phase of the model [see Fig. 1(b)]: when interactions are absent ($V = 0$), the system is in the AFAI phase with $\mu_1 = 2$, while all higher-order invariants take value zero [31]. When interactions are weak, but finite, our numerical results indicate that many-body localization persists, and hence the system remains in the AFI phase with $\mu_1 = 2$ (here the factor of 2 accounts for the two spin species). In particular, the values of all higher-order invariants must remain zero [$S_2 = 0$, see Fig. 1(c)]. However, when interactions are much stronger than the tunneling rate between the sites, $J$, they act to block tunneling to or from doubly-occupied sites, resulting in nonzero values of $C_{ij}$ for such configurations. We demonstrate that this effect drives the model into a CIAFI phase with $\mu_2 = -2$ ($S_2 = -2A/T$). In Fig. 1(d), we confirm that the transition between the AFI and CIAFI phases in this model is accompanied by a divergence of the localization length of the two-particle states of the system.

## II. MANY-BODY AND $k$-PARTICLE LOCALIZATION IN PERIODICALLY DRIVEN SYSTEMS

The main result of this work is to characterize the topological properties of time-evolution in two-dimensional periodically-driven $k$-particle (or many-body) localized fermionic systems. As a preliminary step, in this section we review the structure of the Floquet operator in such systems.

The system we study is a two-dimensional lattice systems of interacting fermions, of physical dimensions $L \times L$, subject to periodic driving. While our results apply to any type of lattice, below we assume for simplicity that the system is defined on a square lattice with lattice constant $a$ and (time-dependent) nearest-neighbor tunneling. The time evolution of the system is described by the time-periodic Hamiltonian $H(t) = H(t + T)$, where $T$ is the driving period. To avoid complications from the coexistence of thermalizing chiral edge states and a localized bulk [37], we focus on the case where the system is defined on a torus, such that no edges are present [44].

### A. Structure of Floquet operator in many-body localized systems

We first review the structure of the Floquet operator when the system is many-body localized, i.e., when any state of the system exhibits localized behavior in the thermodynamic limit. The concepts we introduce here also form a basis for our discussion of the more general case of $k$-particle localization (Sec. II B).

When the system is MBL, it has a complete set of emergent local integrals of motion [39, 40, 45, 46] (LIOMs),

$\{\hat{n}_a\}$. The LIOMs form a mutually commuting set of quasilocal operators that are individually preserved by the stroboscopic evolution of the system [47]. The number of independent LIOMs in the localized system is given by the dimension $D_1$ of the system's single-particle Hilbert space. For spinless fermions with one orbital per site, we have $D_1 = L^2/a^2$. The LIOMs $\{\hat{n}_\alpha\}$ may thus be labelled by a single index $\alpha$ which runs from 1 to $D_1$.

To make the discussion more concrete, the LIOMs can be identified from the system's Floquet operator [39], $U(T)$. The Floquet operator is defined as the evolution operator of the system, $U(t) \equiv \mathcal{T}e^{-i\int_0^t dt\, H(t)}$, evaluated for a time interval corresponding to one complete driving period $T$. Here $\mathcal{T}$ denotes the time-ordering operation, and we work in units where $\hbar = 1$ throughout. Analogously to nondriven systems, the stroboscopic time-evolution (i.e., the time-evolution at integer multiples of the driving period $T$) is conveniently expressed in terms of the eigenstates of the Floquet operator, $\{|\psi_n\rangle\}$, known as *Floquet eigenstates*. These satisfy $U(T)|\psi_n\rangle = e^{-i\varepsilon_n T}$, where $\varepsilon_n$ has units of energy and is known as quasienergy. Note that each quasienergy $\varepsilon_n$ is only defined modulo the driving frequency $\Omega \equiv 2\pi/T$. The stroboscopic time-evolution is hence equivalent to that generated by the static effective Hamiltonian, $H_{\text{eff}} \equiv \sum_n \varepsilon_n |\psi_n\rangle\langle\psi_n|$, since $U(T) = e^{-iH_{\text{eff}}T}$.

In the many-body localized regime, the effective Hamiltonian takes the form

$$H_{\text{eff}} = \sum_{\alpha_1} \varepsilon_{\alpha_1}\hat{n}_{\alpha_1} + \sum_{\alpha_1,\alpha_2} \varepsilon_{\alpha_1\alpha_2}\hat{n}_{\alpha_1}\hat{n}_{\alpha_2} + \cdots . \qquad (1)$$

Each coefficient $\varepsilon_{\alpha_1\ldots a_\ell}$ (referred to as a quasienergy coefficient in the following) is associated with a particular combination $\hat{n}_{\alpha_1}\ldots\hat{n}_{\alpha_k}$ formed from the $D$ distinct LIOMs, and has units of energy. Each sum $\sum_{\alpha_1\ldots\alpha_\ell}$ in Eq. (1) runs over all $\binom{D}{\ell}$ combinations of $\ell$ distinct LIOMs, where $\binom{a}{b}$ denotes the binomial coefficient. The above form of the Floquet operator implies that each LIOM $\hat{n}_\alpha$ is preserved by the stroboscopic evolution of the system, and thus the operators $\{\hat{n}_\alpha\}$ are integrals of motion.

We now review some important properties of the LIOMs which we use in the following. Firstly, each LIOM $\hat{n}_\alpha$ can be written in the form of a fermionic counting operator: $\hat{n}_\alpha = \hat{f}_\alpha^\dagger \hat{f}_\alpha$. Here $\hat{f}_\alpha$ is a (dressed) quasilocal fermionic annihilation operator, constructed from the original lattice annihilation and creation operators $\{\hat{c}_i\}$ and $\{\hat{c}_i^\dagger\}$, respectively, as: $\hat{f}_\alpha = \sum_i \psi_i^\alpha \hat{c}_i + \sum_{ijk} \psi_{ijk}^\alpha \hat{c}_i^\dagger \hat{c}_j \hat{c}_k + \sum_{i\ldots m} \psi_{ijklm}^\alpha \hat{c}_i^\dagger \hat{c}_j^\dagger \hat{c}_k \hat{c}_l \hat{c}_m + \cdots$, where $\hat{c}_i$ annihilates a fermion on site $i$ in the lattice. Through the identification of the LIOMs with fermionic counting operators, we note that $\sum_\alpha \hat{n}_\alpha$ gives the total number of fermions in the system.

Another crucial property of the LIOMs is that each LIOM $\hat{n}_\alpha$ has its support localized around a particular location $\mathbf{r}_\alpha$ in the lattice. Specifically, the magnitude of the coefficient $\psi_{i_1\ldots i_\ell}^\alpha$ decreases exponentially

with the distance $s$ from any of the sites $i_1,\ldots i_\ell$ to $\mathbf{r}_\alpha$: $\psi_{i_1\ldots i_\ell}^\alpha \sim e^{-s/\xi_f}$, where the length scale $\xi_f$ sets the spatial extent of the LIOMs. Similarly to the LIOMs, the quasienergy coefficients $\{\varepsilon_{\alpha_1\ldots\alpha_\ell}\}$ also exhibit localized behavior. Specifically, $\varepsilon_{\alpha_1\ldots\alpha_\ell}$ decays as $e^{-d/\xi_\varepsilon}$, where $d$ is the distance between any two of the LIOM centers $\mathbf{r}_{\alpha_1}\ldots\mathbf{r}_{\alpha_k}$; here $\xi_\varepsilon$ is another localization length scale (not necessarily identical to $\xi_f$, see Ref. 48).

As is evident above, MBL systems may be characterized by several distinct localization lengths [48]. In particular, the LIOM expansion above establishes two length scales, $\xi_f$ and $\xi_\varepsilon$. In the following, we will make use of an additional relevant length scale, $\xi_l$, which characterizes the spread of time-evolved operators.

## B. $k$-particle localization

As we explained in the introduction, the topological classification we develop in this work applies to a more general class of systems than those exhibiting full MBL; specifically, the invariants we identify can be defined for any system that is $k$-particle localized for some nonzero $k$. As defined in the introduction, $k$-particle localization is understood as the situation where all Floquet eigenstates holding $\ell$ particles for $\ell = 1,\ldots k$ are localized. In the remainder of this paper we will make use of similar notation, such that $\ell$ always refers to a specific particle-number sector, while $k$ refers to the "degree of localization" of the system: i.e., $k$ is defined as the integer such that Floquet eigenstates in the system with $k$ or fewer particles are localized, while at least one Floquet eigenstate with $k + 1$ particles is delocalized.

For $k$-particle localized systems, we expect a LIOM decomposition and effective Hamiltonian $H_{\text{eff}}$ as defined in Eq. (1) can be written to describe the evolution in Fock space of up to $k$ particles, with the expansion truncated to $k$th order. Full MBL can be seen as a special case of $k$-particle localization; specifically, MBL can be understood as the $k \to \infty$ limit of $k$-particle localization where the localization length of the truncated LIOM expansion described above remains bounded for all $k$.

## III. TOPOLOGICAL INVARIANTS OF THE TIME EVOLUTION

In this section, as the main result of our work, we characterize the micromotion of $k$-particle localized systems (which includes the case of MBL as described above). We show that such systems may exhibit non-trivial micromotion, featuring steady-state circulating currents at long times. We characterize these circulating currents by analyzing the time-averaged magnetization density operator of the system. From this analysis we identify a set of topological invariants $\mu_1\ldots\mu_k$ that characterize the steady-state circulating currents that the system may support.

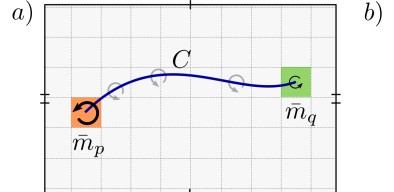 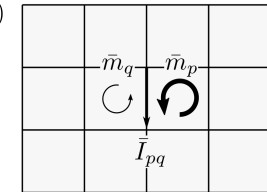

FIG. 2. a) Schematic depiction of the relationship between current and magnetization density [Eq. (4)]. In many-body localized systems, the time-averaged current passing through a cut $C$ is determined by the difference between the currents circulating around the cut's two end-points, $p$ and $q$. The currents circulating around plaquette $p$ are measured by the magnetization density operator $\bar{m}_p$. b) Ampere's law on the lattice. The difference in magnetization densities between two adjacent plaquettes $p$ and $q$ gives the current $\bar{I}_{pq}$ on the bond between them.

In a stepwise fashion, below we consider the dynamics of a $k$-particle localized system in the $\ell$-particle subspace for each $\ell = 1, \ldots k$ (allowing $k$ to be infinite for fully MBL systems). This approach ensures that our our results do not rely on full MBL to be valid, while still applying to this class of systems if such exist.

## A. Characterization of micromotion

To characterize the micromotion of $k$-particle localized systems, in this subsection we consider the dynamics within the subspace of states holding $\ell$ particles, where $\ell \leq k$. Naively, one might expect that the time-averaged current density in this subspace always vanishes due to localization. Indeed, there can be no net flow of charge across any closed curve. However, for an *open* curve (or "cut"), as schematically depicted in Fig. 2a, a nonzero time-averaged current may run across the cut due to uncompensated local circulating currents around the curve's endpoints. The total current circulating around a point in a given plaquette is precisely the magnetization density in this plaquette.

To establish this relationship in more rigorous terms, we consider the total time-averaged current that passes through a cut $C$ between plaquettes $p$ and $q$ in the lattice, as depicted in Fig. 2a. The operator $I_C(t)$ measuring the current through the cut $C$ is given by

$$I_C(t) = \sum_{b \in B_C} I_b(t), \qquad (2)$$

where $I_b$ denotes the bond current operator on bond $b$ (restricted to the $\ell$-particle subspace) [49], and the sum runs over the set $B_C$ of all bonds that cross the cut $C$ [see Appendix A for an explicit definition of $I_b(t)$]. Note that $I_b(t)$, and thereby $I_C(t)$, depends on time in the Schrödinger picture due to the explicit time-dependence of the Hamiltonian $H(t)$.

To characterize the circulating currents in the system, we seek the long-time-averaged expectation value of the current $\langle\langle I_C \rangle\rangle$ for an arbitrary initial $\ell$-particle state, $|\psi\rangle$. Here we introduce the notation $\langle\langle \mathcal{O} \rangle\rangle \equiv \lim_{\tau \to \infty} \frac{1}{\tau} \int_0^\tau dt\, \langle \psi(t)|\mathcal{O}(t)|\psi(t)\rangle$ to indicate the time-averaged expectation value in the state $|\psi(t)\rangle$. The time-averaged current $\langle\langle I_C \rangle\rangle$ may equivalently be computed in the Heisenberg picture as $\langle\langle I_C \rangle\rangle = \langle \psi|\bar{I}_C|\psi\rangle$, where $|\psi\rangle$ denotes the initial many-body state of the system, and $\bar{I}_C$ denotes the long-time-average of the current operator $I_C$ in the Heisenberg picture:

$$\bar{I}_C = \lim_{\tau \to \infty} \frac{1}{\tau} \int_0^\tau dt\, U^\dagger(t) I_C(t) U(t), \qquad (3)$$

where $U(t)$ denotes the system's time-evolution operator as defined above. For later, we define $\overline{\mathcal{O}} \equiv \lim_{\tau \to \infty} \frac{1}{\tau} \int_0^\tau dt\, U^\dagger(t)\mathcal{O}(t)U(t)$ for any operator $\mathcal{O}$.

As argued above, the time-averaged current $\bar{I}_C$ across cut $C$ can only have nonzero expectation value due to localized circulating currents at the cut's two endpoints, $p$ and $q$. This implies that $\bar{I}_C$ only depends on the details of the system near plaquettes $p$ and $q$. In Appendix A we verify this intuition, by proving that the operator $\bar{I}_C$ only has support near the two endpoints of the cut $C$. Specifically, assuming only $k$-particle localization and conservation of charge, we show that, within the $\ell$-particle subspace, where $\ell \leq k$, $\bar{I}_C$ must take the form

$$\bar{I}_C = \bar{m}_p - \bar{m}_q, \qquad (4)$$

where the operator $\bar{m}_p$ has its full support (up to an exponentially small correction) within a distance $\xi_l$ from plaquette $p$, and similarly for $\bar{m}_q$. Here $\xi_l$ is a finite, system-size independent length scale measuring the spread of operators in the system (within the $\ell$-particle subspace): specifically, for any time-periodic operator $A(t)$ with a finite region of support $R$, the long-time average $\bar{A}$ (when restricted to the $\ell$-particle subspace) is a local integral of motion with support within a finite distance $\xi_l$ from $R$ (up to an exponentially small correction) [50].

Crucially, the operator $\bar{m}_p$ in Eq. (4) is the same for *any* cut with an endpoint in plaquette $p$. Thus, Eq. (4) uniquely defines the operator $\bar{m}_p$ for each plaquette $p$ in the system, up to a correction exponentially small in system size. Specifically, let plaquette $q$ be separated from plaquette $p$ by a distance $d$, of order the system size, $L$. In this case, $\bar{m}_p$ can be identified uniquely from the terms of $\bar{I}_C$ which have support nearest to plaquette $p$, up to a correction of order $\mathcal{O}(e^{-d/\xi_l}) \sim \mathcal{O}(e^{-L/\xi_l})$.

For each plaquette $p$, $\bar{m}_p$ may be defined from Eq. (4) as described above by considering a cut of length $\sim L$ (up to an exponentially small correction). The set of operators $\{\bar{m}_p\}$ obtained in this way then obey Eq. (4) for *any* two plaquettes in the lattice. In particular, when the plaquettes $p$ and $q$ are adjacent, Eq. (4) implies that $\bar{m}_p - \bar{m}_q = \bar{I}_{pq}$, where $\bar{I}_{pq}$ measures the time-averaged current on the bond separating plaquettes $p$ and $q$, as schematically depicted in Fig. 2b. This relationship is

the time-averaged lattice version of Ampere's law, which relates the current density, $\mathbf{j}$, to the magnetization density, $\mathbf{m}$: $\mathbf{j} = \nabla \times \mathbf{m}$ (see Ref. 30). We thus identify the operator $\bar{m}_p$ as the time-averaged magnetization density in the system at plaquette $p$ [51]. As the above discussion shows, the time-averaged magnetization $\bar{m}_p$ measures the total current circulating around plaquette $p$.

## B.  Topological invariance of $\mathrm{Tr}_k \, \bar{m}_p$

We now show that, for each value of $\ell = 1, \ldots k$, the trace of $\bar{m}_p$ in the $\ell$-particle subspace, $\mathrm{Tr}_\ell \bar{m}_p$, takes the same value for all plaquettes in the system. Subsequently (in Sec. III B 1) we show that this universal value is quantized as an integer multiple of $1/T$, $z_\ell$. Periodically driven $k$-particle localized systems of fermions in two dimensions are thus characterized by the $k$ integer-valued topological invariants $z_1 \ldots z_k$.

We prove the topological invariance of $\mathrm{Tr}_\ell \, \bar{m}_p$ through a simple line of arguments. First, Eq. (4) implies:

$$\mathrm{Tr}_\ell \, \bar{m}_p - \mathrm{Tr}_\ell \, \bar{m}_q = \mathrm{Tr}_\ell \, \bar{I}_C. \qquad (5)$$

Using the cyclic property of the trace and $U(t)U^\dagger(t) = \mathbf{1}$, we find $\mathrm{Tr}_\ell \, \bar{I}_C = \lim_{\tau \to \infty} \frac{1}{\tau} \int_0^\tau dt \, \mathrm{Tr}_\ell \, I_C(t)$. Recall from Eq. (2) that the current operator $I_C(t)$ is given by a sum of bond current operators. Noting that any bond current operator $I_b(t)$ is by construction traceless (see Appendix A), we conclude that $\mathrm{Tr}_\ell \, \bar{I}_C = 0$. Hence we find:

$$\mathrm{Tr}_\ell \, \bar{m}_p = \mathrm{Tr}_\ell \, \bar{m}_q. \qquad (6)$$

This relation holds for *any* pair of plaquettes in the lattice. Therefore, for a given disorder realization, $\mathrm{Tr}_\ell \, m_p$ must take the same universal value for all plaquettes in the system.

We now show that the universal value of $\mathrm{Tr}_\ell \, \bar{m}_p$ is a topological invariant of the system in the thermodynamic limit ($L \to \infty$) [52]. Consider perturbing $H(t)$ within some subregion $R$ of the system (by a small but finite amount), in such a way that $\ell$-particle localization is preserved. Before and after the perturbation, $\mathrm{Tr}_\ell \, \bar{m}_p$ only depends on the details of the system around the plaquette $p$, up to an exponentially small correction (due to the exponentially decaying tails of the LIOMs). Hence, for a plaquette $p$ located a distance of order $L/2$ from the region $R$, $\mathrm{Tr}_\ell \, \bar{m}_{p_0}$ may only change by an amount of order $e^{-L/2\xi_l}$ due to the perturbation. Since $\mathrm{Tr}_\ell \, \bar{m}_p$ is given by the same value for *all* plaquettes in the system, $\mathrm{Tr}_\ell \, \bar{m}_p$ must remain unaffected by the perturbation even for plaquettes within the region where the system is perturbed, $R$. Thus, $\mathrm{Tr}_\ell \, \bar{m}_p$ is unaffected by any local perturbation that preserves $\ell$-particle localization, up to a correction exponentially suppressed in system size. We conclude that $\mathrm{Tr}_\ell \, \bar{m}_p$ is a topological invariant of the system, protected by $\ell$-particle localization.

In the following, it is convenient to parameterize the topologically-invariant value of $\mathrm{Tr}_\ell \, \bar{m}_p$ by a dimensionless

number; we hence let $z_\ell$ denote the value of $\mathrm{Tr}_\ell \, \bar{m}_p$ in units of the inverse driving period, such that $\mathrm{Tr}_\ell \, \bar{m}_p = z_\ell/T$.

### 1.  Quantization of $z_\ell$

Here we show that the dimensionless invariant $z_\ell$ must take an integer value for each $\ell$. To do this, we use an approach that generalizes the one employed for the non-interacting case in Ref. 30. This subsection provides a summary of the proof, while full details are given in Appendix B.

To begin, we consider the total time-averaged magnetization operator, $\bar{M} \equiv \sum_p \bar{m}_p a^2$. Since $\mathrm{Tr}_\ell \, \bar{m}_p$ takes the value $z_\ell/T$ for all plaquettes in the system, we have

$$\mathrm{Tr}_\ell \bar{M} = z_\ell \, L^2/T. \qquad (7)$$

To establish the quantization of $z_\ell$, we proceed in two steps. First, we obtain $\mathrm{Tr}_\ell \bar{M}$ from the response of the system to the insertion of the weak uniform magnetic field $B_0 = 2\pi/L^2$ that corresponds to one flux quantum piercing the torus (note that the flux quantum is given by $2\pi$ in the units we employ): we show that, in the thermodynamic limit,

$$e^{-i\mathrm{Tr}_\ell(\bar{M})B_0 T} = |\tilde{U}(T)|_\ell/|U(T)|_\ell, \qquad (8)$$

where $\tilde{U}(T)$ denotes the Floquet operator of the system in the presence of the magnetic field $B_0$, and $| \cdot |_\ell$ denotes the determinant within the $\ell$-particle subspace. Subsequently, we show that the determinants $|\tilde{U}|_\ell$ and $|U|_\ell$ must be identical (see also Ref. 30); this implies that $\mathrm{Tr}_\ell(\bar{M})B_0 T$ equals an integer multiple of $2\pi$. Using $B_0 = 2\pi/L^2$ along with Eq. (7), we conclude that $z_\ell$ must be an integer.

To obtain Eq. (8) (which forms the first step in our derivation), we show that the magnetic moment of each $\ell$-particle Floquet eigenstate, $|\psi_n\rangle$, gives the response of its quasienergy, $\varepsilon_n$, to the addition of the weak magnetic field $B_0$. Letting $\tilde{\varepsilon}_n$ denote the perturbed quasienergy level in the one-flux system associated with $|\psi_n\rangle$ (see the following for details, and, in particular, for a discussion of the perturbation-induced resonances), we show in Appendix B that

$$\tilde{\varepsilon}_n - \varepsilon_n \approx -\langle\psi_n|\bar{M}|\psi_n\rangle B_0. \qquad (9)$$

Specifically, the sum of $\tilde{\varepsilon}_n - \varepsilon_n$ over *all* $\ell$-particle Floquet states satisfies

$$\sum_n (\tilde{\varepsilon}_n - \varepsilon_n) = -\sum_n \langle\psi_n|\bar{M}|\psi_n\rangle B_0 + \mathcal{O}(e^{-L/\xi}), \quad (10)$$

where $\mathcal{O}(e^{-L/\xi})$ denotes some (dimensionfull) correction which goes to zero as $e^{-L/\xi}$ in the thermodynamic limit. We obtain Eq. (8) from Eq. (10) by multiplying with $-iT$, taking the exponentials on both sides and recalling that $|\tilde{U}(T)|_\ell = \exp(-i\sum_n \tilde{\varepsilon}_n T)$ and likewise for $U(T)$.

Eq. (10) can be obtained through first-order perturbation theory in $B_0$. In Appendix B, we provide a rigorous derivation of this result, along with an exact definition of the one-to-one relationship between the quasienergy levels of the one- and zero-flux systems which Eq. (10) implicitly requires. (In particular, we give the prescription for uniquely identifying $\tilde{\varepsilon}_n$ for each "unperturbed" quasienergy level $\varepsilon_n$.). Here we summarize the arguments: near the region of support of $|\psi_n\rangle$ [53], the Hamiltonian of the one-flux system, $\tilde{H}(t)$, is given by $H(t) - \sum_b I_b(t)\theta_b + \mathcal{O}(\theta_b^2)$, where $\theta_b$ denotes the Peierls phase on bond $b$ induced by the magnetic field $B_0$, and $I_b(t)$ denotes the bond current operator (see Sec. III A and Appendix A). Note that there is a gauge freedom in choosing the Peierls phases; we choose them to be of order $1/L^2$ near the region of support of $|\psi_n\rangle$ (such that the subleading correction in the above expansion of $\tilde{H}(t)$ can be neglected in the thermodynamic limit).

In the thermodynamic limit $L \to \infty$, one may naively expect that the quasienergy spectrum of the one-flux system can be obtained through a first-order perturbative expansion in $\delta H(t) \equiv \tilde{H}(t) - H(t)$ (for each $|\psi_n\rangle$), which is approximately identical to $-\sum_b I_b(t)\theta_b$. However, note that the convergence of such an expansion to first order is only ensured if the ratio between the matrix elements of $\delta H$ in the Floquet eigenstate basis and the corresponding quasienergy level spacings, $r_{mn} \equiv \langle\psi_m|\delta H(t)|\psi_n\rangle/(\varepsilon_m - \varepsilon_n)$, is much smaller than 1 for *all* choices of $\ell$-particle Floquet eigenstates $m$ and $n$. While the perturbation $\delta H(t)$ is of order $L^{-2}$, the many-body level spacing in the $\ell$-particle subspace is of order $\Omega/(L^{2\ell})$, where $\Omega \equiv 2\pi/T$ denotes the angular driving frequency. Hence, in the thermodynamic limit $r_{mn}$ can potentially be much larger than 1 for certain choices of $m$ and $n$. However, in Appendix B we provide a careful analysis that confirms our initial expectation: with a probability that goes to 1 in the thermodynamic limit (for each $\ell$ between 1 and $k$), $r_{nm}$ goes to zero for *all* choices of $m$ and $n$. This result arises because states where $\langle\psi_n|\delta H|\psi_m\rangle$ is nonvanishing must be spatially close, and hence experience local level repulsion.

The above discussion shows that the quasienergy level corresponding to the state $|\psi_n\rangle$ in the one-flux system, $\tilde{\varepsilon}_n$, is captured by first-order perturbation theory with respect to $\delta H(t)$. Expanding the quasienergy $\tilde{\varepsilon}_n$ to first order in $\delta H(T)$, we obtain

$$\tilde{\varepsilon}_n - \varepsilon_n \approx \frac{1}{T}\int_0^T dt\, \langle\psi_n|U^\dagger(t)\delta H(t)U(t)|\psi_n\rangle \qquad (11)$$

(see Appendix B for proof). Using $\delta H(t) \approx -\sum_b \theta_b I_b(t)$ along with the fact that in a Floquet eigenstate the time-averaged expectation value over one period is identical to the long-time average, we find

$$\tilde{\varepsilon}_n - \varepsilon_n \approx -\sum_b \theta_b \langle\psi_n|\bar{I}_b|\psi_n\rangle, \qquad (12)$$

where $\bar{I}_b$ denotes the long-time average of the bond current $I_b(t)$ in the Heisenberg picture (see Sec. III A).

Recall from Eq. (4) (see also Fig. 2b) that $\bar{I}_b = \bar{m}_{p_b} - \bar{m}_{q_b}$, where $p_b$ and $q_b$ denotes the two adjacent plaquettes separated by the bond $b$, such that $b$ is oriented counterclockwise with respect to $p_b$ [49]. Inserting this result into Eq. (12), we note that each plaquette in the lattice appears four times exactly (namely once for each of the four bonds bounding the plaquette). Rearranging the terms from a sum over bonds to a sum over plaquettes, we thus find

$$\tilde{\varepsilon}_n - \varepsilon_n \approx -\sum_p \langle\psi_n|\bar{m}_p|\psi_n\rangle(\theta_{b_{p,1}} + \theta_{b_{p,2}} + \theta_{b_{p,3}} + \theta_{b_{p,4}}).$$
$$(13)$$

where $b_{p,i}$ denotes the lattice bond that constitutes the $i$th edge of plaquette $p$ (counted in clockwise order starting from the positive $x$-direction), and $\theta_{b_{p_i}}$ gives the Peierls phase acquired by traversing the bond counterclockwise with respect to $p$. The sum of Peierls phases $\theta_{b_{p,1}} + \theta_{b_{p,2}} + \theta_{b_{p,3}} + \theta_{b_{p,4}}$ hence gives the flux through plaquette $p$, and hence yields exactly $B_0 a^2$ for each plaquette. Eq. (9) follows by using $\bar{M} \equiv \sum_p a^2 \bar{m}_p$.

The rigorous derivation in Appendix B shows that the correction to the approximate equality in Eq. (9) scales with system size as $L^{-4}$, and hence is subleading in thermodynamic limit (recall that $B_0 \sim L^2$). We subsequently use the LIOM structure of the Floquet operator in Eq. (1) to show that, remarkably, these individual corrections approximately cancel out when summed over all $\ell$-particle states, yielding an *exponentially* suppressed net correction, which scales with system size as $e^{-L/\xi}$. This establishes Eq. (10), and thereby also Eq. (8).

What remains to be shown is that $U(T)$ and $\tilde{U}(T)$ have identical determinants in the $\ell$-particle subspace. We show this using the approach from Ref. 30: the determinant of any time-evolution operator can be found from the time-integrated trace of the Hamiltonian [17]: $|U(T)|_\ell = e^{-i\int_0^T dt'\, \mathrm{Tr}_\ell H(t)}$. This follows because

$$\sum_n \varepsilon_n = -\frac{i}{T}\int_0^T dt\, \mathrm{Tr}[U^\dagger(t)\partial_t U(t)]_\ell, \qquad (14)$$

which can be straightforwardly verified using the spectral decomposition of $U(t)$. Identifying the integrand in the right-hand side above as $-i\mathrm{Tr}[H(t)]_\ell$, we find $|U(T)|_\ell = \exp(-i\int_0^T dt\, \mathrm{Tr}[H(t)]_\ell)$. Since the insertion of a magnetic flux only modifies off-diagonal elements of the Hamiltonian (in the lattice site basis), the trace of the Hamiltonian is unaffected by the magnetic field $B_0$. Thus $|\tilde{U}(T)|_\ell = |U(T)|_\ell$. Hence, the right-hand side of Eq. (8) equals 1 and therefore the argument in the exponent of $e^{-i\mathrm{Tr}_\ell(\bar{M})B_0 T}$ must be an integer multiple of $2\pi$. Combining this with Eq. (7) and using that $B_0 = 2\pi/L^2$, we conclude that $z_\ell$ must be an integer.

## C. Cumulant basis of invariants

The above discussion shows that $k$-particle localized systems are characterized by the $k$ independent, integer-valued topological invariants $z_1 \ldots z_k$. Here $z_\ell$ gives the trace of the magnetization density operator in the $\ell$-particle subspace (in units of the inverse driving period). However, each $z_\ell$ depends on the size of the system, and thus is not an *intrinsic* property of the system. For instance, in noninteracting systems, $z_\ell$ scales as $L^{2(\ell-1)}$, where $L$ is the physical dimension of the system [54]. In this subsection we construct linear combinations of the invariants $z_1 \ldots z_k$ that give an equivalent set of system size *independent* invariants $\mu_1 \ldots \mu_k$ that characterize the *intrinsic* topological properties of the system.

The intrinsic invariants $\mu_1 \ldots \mu_k$ can be expressed as the cumulants of the magnetization operator, as discussed in Sec. I. To illustrate, consider the time-averaged magnetic moment, $\bar{M} \equiv \sum_p a^2 \bar{m}_p$, of a state where two particles are initialized on sites $i$ and $j$, which we denote $\bar{M}_{ij}$. The average of the total magnetic moment, taken over *all* 2-particle states, is given by $\frac{1}{D_2}(z_2 L^2/T)$, where $D_\ell$ denotes the dimension of the $\ell$-particle subspace. For each $i$ and $j$, we write $\bar{M}_{ij} = \bar{M}_i + \bar{M}_j + C_{ij}$, where, as in Sec. I, $\bar{M}_i$ denotes the time-averaged magnetization of the system holding a single particle initially located at site $i$. From this definition of $C_{ij}$, we find

$$\frac{1}{L^2} \sum_{i<j} C_{ij} = \frac{z_2 - 2(L^2-1)z_1}{T}, \qquad (15)$$

where we used that $\mathrm{Tr}_\ell \bar{M} = z_\ell L^2/T$ for $\ell = 1, 2$. The right hand side is evidently an integer multiple of $1/T$. We take this integer to be our definition of the intrinsic invariant $\mu_2$.

Note that $\mu_2$ gives the mean value of $S_i \equiv \sum_{j \neq i} C_{ij}$ over all sites $i$ (recall that $C_{ij} = C_{ji}$). Importantly, due to the fact that the two particles only influence each other's motion when they are within a localization length of one another, the cumulant $C_{ij}$ is only significant for $\mathcal{O}(\xi_l^2/a^2)$ choices of $j$ for each $i$. The mean value of $S_i$ is therefore an intrinsic quantity, which does not depend on the system size; in particular, it remains finite in the thermodynamic limit. In the noninteracting case, $C_{ij} = 0$, and $\mu_2 = 0$. Thus, $\mu_2$ gives the contribution to the magnetization from 2-particle correlations.

We extend this definition to higher numbers of particles, by expanding $\bar{M}$ in terms of the fermionic annihilation and creation operators, $\{\hat{c}_i\}, \{\hat{c}_i^\dagger\}$. Since $\bar{M}$ preserves the number of particles, we have

$$\bar{M} = \sum_{i_1 j_1} \mathcal{M}_{i_1;j_1} \hat{c}_{i_1}^\dagger \hat{c}_{j_1} + \sum_{i_1 i_2, j_1 j_2} \mathcal{M}_{i_1 i_2; j_1 j_2} \hat{c}_{i_1}^\dagger \hat{c}_{i_2}^\dagger \hat{c}_{j_1} \hat{c}_{j_2} + \cdots. \qquad (16)$$

Without loss of generality, we take $\mathcal{M}_{i_1 \ldots i_k; j_1 \ldots j_k}$ to be nonzero only if $i_1 < i_2 \ldots < i_k$ and $j_1 > j_2 \ldots > j_k$, such that each independent combination of creation and annihilation operators appears only once in the above

sum. We see that the expectation value of $\bar{M}$ in a single-particle state $|i\rangle \equiv \hat{c}_i^\dagger |0\rangle$ (where $|0\rangle$ denotes the vacuum state) is given by $\mathcal{M}_{i;i}$. We thus identify $\mathcal{M}_{ii} = \bar{M}_i$, where $\bar{M}_i$ was defined above. Likewise, in the two-particle-state $|ij\rangle \equiv \hat{c}_i^\dagger \hat{c}_j^\dagger |0\rangle$ (where $i < j$), the expectation value of $\bar{M}$ is given by $\mathcal{M}_{i;i} + \mathcal{M}_{j;j} + \mathcal{M}_{ij;ji}$. We thus identify $\mathcal{M}_{ij;ji} = C_{ij}$. The higher-order cumulants can be defined in a similar fashion, such that $C_{i_1, \ldots i_\ell} = \mathcal{M}_{i_1 \ldots i_\ell; i_\ell \ldots i_1}$. Note that the long-time average of an operator in the Heisenberg picture, such as $\bar{M}$, must be diagonal in the Floquet eigenstate basis; for example, $\mathcal{M}_{i;j}$ is diagonal in the basis of single-particle Floquet eigenstates.

Due to localization and the locality of interactions (see above), the coefficient $C_{i_1 \ldots i_\ell}$ can only be nonzero if all sites $i_1 \ldots i_\ell$ are spatially close (on the scale of $\xi_l$). Thus, through arguments analogous to those below Eq. (15), for each $\ell = 1 \ldots k$, $\frac{T}{L^2} \sum_{i_1, \ldots i_\ell} C_{i_1 \ldots i_\ell}$ is a (dimensionless) intrinsic quantity of the system. This motivates us to define the $\ell$-th intrinsic invariant as:

$$\mu_\ell = \frac{T}{L^2} \sum_{i_1 \ldots i_\ell} C_{i_1 \ldots i_\ell}. \qquad (17)$$

To relate $\mu_\ell$ to the invariants $z_1 \ldots z_k$, we take the $\ell$-particle trace in Eq. (16). Using $\mathrm{Tr}_\ell[\hat{c}_{i_1}^\dagger \ldots \hat{c}_{i_\nu}^\dagger \hat{c}_{i_\nu} \ldots \hat{c}_{i_1}] = \binom{D_1 - \nu}{\ell - \nu}$ (this can be verified from combinatorial arguments), where $D_1 = L^2$ denotes the dimension of the system's single-particle subspace, we find

$$z_\ell = \sum_{\nu=1}^{\ell} \binom{D_1 - \nu}{\ell - \nu} \mu_\nu, \qquad (18)$$

where we used $\mathrm{Tr}_\ell \bar{M} = z_\ell L^2/T$. By induction, one can verify that each $\mu_\ell$ is an integer. First, by the definition above, $\mu_1$ equals $z_1$, and hence is an integer. For $\ell > 1$, $\mu_\ell = z_\ell - \sum_{\nu=1}^{\ell-1} \binom{D_1-\nu}{\ell-\nu} \mu_\nu$. Thus, if $\mu_1 \ldots \mu_{\ell-1}$ are integers, $\mu_\ell$ is also an integer (since $z_\ell$ is an integer).

To further elucidate the physical meaning of the intrinsic invariant $\mu_\ell$, we express it in terms of the LIOMs that were introduced in Sec. II. Since the long-time average of any Heisenberg picture operator is diagonal in the basis of Floquet eigenstates [55], the operator $\bar{m}_p$ must be an integral of motion [56]. This requires $\bar{m}_p$ to take the following form in terms of the of the LIOMs $\{\hat{n}_\alpha\}$ that we introduced in Eq. (1):

$$\bar{m}_p = \sum_{\alpha_1} m_{\alpha_1}^p \hat{n}_{\alpha_1} + \sum_{\alpha_1 \alpha_2} m_{\alpha_1 \alpha_2}^p \hat{n}_{\alpha_1} \hat{n}_{\alpha_2} + \cdots. \qquad (19)$$

Here, for each term involving a products of $\ell$ LIOMs, the sum $\sum_{\alpha_1 \ldots \alpha_\ell}$ runs over the $\binom{D_1}{\ell}$ distinct combinations of $\ell$ LIOM indices $\alpha_1 \ldots \alpha_\ell$. Due to the finite support of the operator $\bar{m}_p$, we note that the coefficient $m_{\alpha_1 \ldots \alpha_\ell}^p$ vanishes as $e^{-d/\xi_l}$, where $d$ is the distance from the plaquette $p$ to the center of the most remote of the LIOMs $\alpha_1 \ldots \alpha_\ell$.

Taking the $\ell$-particle trace in Eq. (19) and using $\text{Tr}_\ell[\hat{n}_{\alpha_1}\ldots\hat{n}_{\alpha_\nu}] = \binom{D-\nu}{\ell-\nu}$, we find

$$z_\ell = \sum_{\nu=1}^{\ell} \binom{D_1-\nu}{\ell-\nu} \sum_{\alpha_1\ldots\alpha_\nu} m^p_{\alpha_1\ldots\alpha_\nu}/T. \qquad (20)$$

Comparing with Eq. (18) for each $\ell = 1\ldots k$, we find

$$\mu_\ell \equiv \sum_{\alpha_1\ldots\alpha_\ell} m^p_{\alpha_1\ldots\alpha_\ell}/T. \qquad (21)$$

Note that $\mu_\ell$ is independent of the choice of plaquette $p$.

From the expression above, it is evident that $\mu_\ell$ characterizes the intrinsic topological properties of the system. Since the magnetization coefficients $\{m^p_{\alpha_1\ldots\alpha_\ell}\}$ vanish when the distance from any of the LIOM centers $\mathbf{r}_{\alpha_1}\ldots\mathbf{r}_{\alpha_\ell}$ to plaquette $p$ becomes large, the right-hand side of Eq. (21) is independent of system size in the thermodynamic limit. In essence, $\mu_\ell$ captures the contribution of $\ell$-body correlations to the magnetization density.

### D. Quantized magnetization density in fully occupied regions

As a final part of this section, we show that the values of the invariants $\mu_1\ldots\mu_k$ can be measured directly from the magnetization density within a region of the system where all sites are occupied. In particular, for the AFI (which is fully MBL and for which only $\mu_1$ takes nonzero value), the magnetization density is given by $\mu_1/T$.

Consider preparing the system in an $\ell$-particle state $|\Psi_{\mathcal{R}}\rangle$ (where $\ell \leq k$) by filling all sites in some finite region of the lattice, $\mathcal{R}$, of linear dimension $d$, with all sites outside $\mathcal{R}$ remaining empty (here we assume this requires fewer than $k$ particles). For a plaquette $p$ located deep within the fully occupied region, we find the time-averaged magnetization density as $\langle\!\langle m_p\rangle\!\rangle = \langle \bar{m}_p\rangle_{\mathcal{R}}$, where we introduced the shorthand $\langle\mathcal{O}\rangle_{\mathcal{R}} \equiv \langle\Psi_{\mathcal{R}}|\mathcal{O}|\Psi_{\mathcal{R}}\rangle$. Using the expansion of $\bar{m}_p$ in Eq. (19), we thus find:

$$\langle\!\langle m_p\rangle\!\rangle = \sum_{\alpha_1} m^p_{\alpha_1}\langle\hat{n}_{\alpha_1}\rangle_{\mathcal{R}} + \sum_{\alpha_1\alpha_2} m^p_{\alpha_1\alpha_2}\langle\hat{n}_{\alpha_1}\hat{n}_{\alpha_2}\rangle_{\mathcal{R}} + \cdots. \qquad (22)$$

To analyze the sum, we note that, for a LIOM $\hat{n}_a$ whose center $\mathbf{r}_a$ is located deep within the filled region $\mathcal{R}$, all sites where $\hat{n}_a$ has its support are occupied. Thus $\hat{n}_\alpha|\Psi_{\mathcal{R}}\rangle = |\Psi_{\mathcal{R}}\rangle + \mathcal{O}(e^{-d/\xi_l})$ [57]. Here the correction arises from the exponentially decaying tail of $\hat{n}_\alpha$ outside the filled region. For terms in the above equation where the centers of all the LIOMs $\alpha_1\ldots\alpha_\nu$ are located near the plaquette $p$, the above result implies that $\langle\hat{n}_{\alpha_1}\ldots\hat{n}_{\alpha_\nu}\rangle_{\mathcal{R}} = 1 + \mathcal{O}(e^{-d/\xi_l})$, since *all* of the LIOMs $\hat{n}_{\alpha_1}\ldots\hat{n}_{\alpha_\nu}$ are located deep within the initially occupied region. For all remaining terms in Eq. (22), one or more LIOMs $\alpha_1\ldots\alpha_\nu$ are located outside the filled region, and thus reside at least a distance $\sim d$ from the plaquette $p$. In this case, the coefficient $m^p_{\alpha_1\ldots\alpha_\nu}$

is exponentially small in $d/\xi_l$ [see the discussion below Eq. (19)]. For both categories of terms we can thus set $\langle\Psi_{\mathcal{R}}|m^p_{\alpha_1\ldots\alpha_\nu}\hat{n}_{\alpha_1}\ldots\hat{n}_{\alpha_\nu}|\Psi_{\mathcal{R}}\rangle = m^p_{\alpha_1\ldots\alpha_\nu}$, at the cost of a correction of order $e^{-d/\xi_l}$. Doing so, we obtain

$$\langle\!\langle m_p\rangle\!\rangle = \sum_{\alpha_1} m^p_{\alpha_1} + \sum_{\alpha_1\alpha_2} m^p_{\alpha_1\alpha_2} + \ldots + \mathcal{O}(e^{-d/\xi_l}).$$

Using Eq. (21), we identify the $\ell$-th sum above as the invariant $\mu_\ell/T$. Recalling that $\langle\Psi_{\mathcal{R}}|\bar{m}_p|\Psi_{\mathcal{R}}\rangle = \langle\!\langle m_p\rangle\!\rangle$, we thus find:

$$\langle\!\langle m_p\rangle\!\rangle = \frac{1}{T}\sum_{\nu=1}^{\ell}\mu_\nu + \mathcal{O}(e^{-d/\xi_l}). \qquad (23)$$

The above discussion thus shows that the magnetization density deep within the filled region is given by the (convergent [58]) sum of the invariants $\{\mu_\ell\}$. In particular, for the AFI, where only $\mu_1$ is nonzero, $\langle\!\langle m_p\rangle\!\rangle = \mu_1/T$.

We note that the individual invariants $\mu_1\ldots\mu_k$ may be extracted from the dependence of the magnetization density on the particle density in the system. Specifically, for a random initial state with a uniform, finite particle density $\rho$, the expectation value $\langle\hat{n}_{\alpha_1}\ldots\hat{n}_{\alpha_\nu}\rangle$, averaged over all choices of LIOMs, is given by $\rho^\nu$. Hence, at finite particle density $\rho$, the average magnetization density in the system is given by $\langle\!\langle m_p\rangle\!\rangle \approx \frac{1}{T}\sum_{\nu=1}^{\ell}\mu_\nu\rho^\nu$. The values of the individual invariants $\mu_\nu$ can thus be extracted from a fit of $\langle\!\langle m_p\rangle\!\rangle$ as a function of $\rho$.

### IV. SPECIFIC MODEL AND NUMERICAL SIMULATIONS

In this section we present a simple model for a periodically driven system of interacting fermions in two dimensions, which realizes either the AFI or a CIAFI phase. The model was briefly discussed in Sec. I. We first consider the limit of weak interaction. In this regime we argue that the system realizes the AFI phase with $\mu_1 = 2$. Subsequently, we show that, in the limit of strong interactions, the model is characterized by a quantized, nonzero value of the "two-particle cumulant" of the magnetization density, consistently with a CIAFI phase characterized by $\mu_2 = -2$. To support our conclusions, we provide numerical simulations of the model in the to above regimes.

The model we consider consists of fermions with spin-1/2 living in a two-dimensional bipartite square lattice with periodic boundary conditions. The Hamiltonian is given by

$$H(t) = H_{\text{dr}}(t) + H_{\text{dis}} + H_{\text{int}}, \qquad (24)$$

where $H_{\text{dr}}(t)$ describes piecewise-constant, time-dependent hopping, $H_{\text{dis}}$ denotes a disorder potential, while $H_{\text{int}}$ describes an on-site interaction between the fermions. The driving protocol, which is contained in $H_{\text{dr}}(t)$, is divided into five segments, as depicted in Fig. 2a. The first four segments each have duration

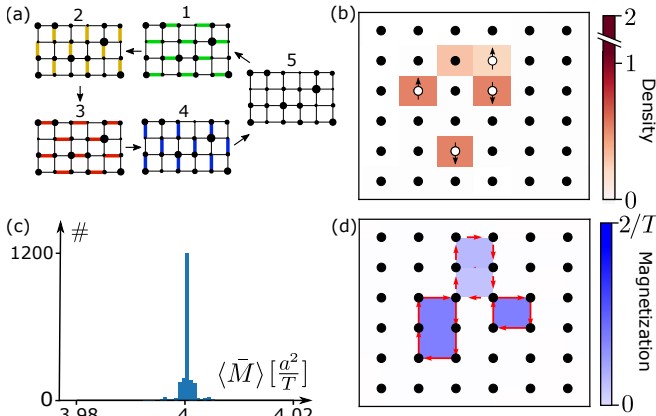

FIG. 3. Simulations of model studied in Sec. IV in the case of weak interactions, where it realizes the AFI phase (see main text for details). (a) Schematic depiction of the driving protocol. (b) Final particle density after 1000 driving periods for initialization in a random on-site configuration of particles (initial configuration of particles marked by white). (c) Histogram of magnetic moments of 246 randomly chosen initial states that are evolved for 1000 periods (see main text for details). A single outlier at value 3.89 is not shown here. (d) Time-averaged bond-current (red) and magnetization density in the system (blue) for the realization depicted in panel (a).

$\eta T/4$, while the fifth segment has duration $(1-\eta)T$; the parameter $\eta$ is a number between 0 and 1 which controls the localization properties of the model (see below). In the first four segments, $H_{\mathrm{dr}}(t)$ turns hopping on for the four different bond types in a counterclockwise fashion, as indicated in Fig. 3a, while $H_{\mathrm{dr}}(t) = 0$ in the fifth segment. More specifically, in the $j$-th segment (where $j \leq 4$),

$$H_{\mathrm{dr}}(t) = J \sum_{\mathbf{r} \in A} \sum_{s} (\hat{c}^{\dagger}_{\mathbf{r}+\mathbf{b}_j,s}\hat{c}_{\mathbf{r},s} + \text{h.c.}). \qquad (25)$$

Here $\hat{c}_{\mathbf{r},s}$ annihilates a fermion on site $\mathbf{r}$ with spin $s$, and the vectors $\{\mathbf{b}_j\}$ are given by $\mathbf{b}_1 = -\mathbf{b}_3 = (a,0)$ and $\mathbf{b}_2 = -\mathbf{b}_4 = (0,a)$. The $\mathbf{r}$-sum above runs over all sites in sublattice A of the bipartite square lattice. We set the tunneling strength to $J = \frac{2\pi}{\eta T}$, such that, in the absence of disorder and interactions, $H_{\mathrm{dr}}$ would generate a perfect transfer of particles across the active bonds in each of the first four segments. The parameter $\eta$ controls how rapidly the "hopping $\pi$-pulses" are applied (and thereby how strong they are relative to the disorder and interaction potentials), and thus controls the localization properties of the model; smaller $\eta$ yields stronger localization (see Ref. 37).

The disorder and interaction terms $H_{\mathrm{dis}}$ and $H_{\mathrm{int}}$ are constant throughout the driving period and are given by

$$H_{\mathrm{dis}} = \sum_{\mathbf{r},s} w_{\mathbf{r}}\hat{\rho}_{\mathbf{r},s}, \quad H_{\mathrm{int}} = V \sum_{\mathbf{r}} \hat{\rho}_{\mathbf{r},\uparrow}\hat{\rho}_{\mathbf{r},\downarrow}. \qquad (26)$$

For each site, $w_{\mathbf{r}}$ takes a random value in the interval $[-W, W]$, and $\hat{\rho}_{\mathbf{r},s} \equiv \hat{c}^{\dagger}_{\mathbf{r},s}\hat{c}_{\mathbf{r},s}$ denotes the occupancy on

site $\mathbf{r}$. The parameter $V$ has units of energy and denotes the strength of the interactions. Note that when $V \gg J$, tunneling is effectively blocked between doubly-occupied and vacant sites. As we show below, this blocking leads to a nonzero value of the higher-order invariant $\mu_2$.

To characterize the topological properties of the model, we consider the dynamics of particles in the two limits of weak and strong interactions. Below we demonstrate how these two regimes drive the model into the AFI phase with $\mu_2 = 2$ and a CIAFI phase with $\mu_2 = -2$, respectively. We substantiate these conclusions with numerical simulations in Sec. IV A.

In the absence of interactions, $V = 0$, the model in Eq. (24) reduces to two decoupled copies of the AFAI model from Ref. 31. When interactions are weak, but nonzero, Ref. 37 suggests that the phase remains MBL (i.e., non-thermalizing). Since the model should be connected to the non-interacting AFAI, we hence expect the system to be in the AFI phase [37] with winding number $\mu_1 = 2$ (see also discussion in Sec. I). The factor of 2 arises from the extra species of fermions introduced due to the spin-1/2 degree of freedom.

We now show that the model above is in a CIAFI phase with $\mu_2 = -2$ in the limit of strong interactions, $V \to \infty$. To see this, we consider the time-averaged magnetic moment $\bar{M}_{ij}$ (see Sec. III C) that results when initially occupying two single-particle states $i$ and $j$, where each choice of $i$ or $j$ corresponds to a particular site and spin. Recall that tunneling is blocked when the first particle is located on, or tunnels to, a site occupied by the second particle. Hence, doublons (i.e., states where two particles occupy the same site) remain frozen in place, implying that $\bar{M}_{ij} = 0$ if $i$ and $j$ correspond to the same site being occupied. For all other initial configurations, interactions effectively do not affect the dynamics, and one can verify that $\bar{M}_{ij} = \bar{M}_i + \bar{M}_j$, where $\bar{M}_i$ denotes the time-averaged magnetic moment in the single-particle state $i$. As a result, the "cumulant" $C_{ij} \equiv \bar{M}_{ij} - \bar{M}_i - \bar{M}_j$ takes value $-2a^2/T$ when the initialization $ij$ corresponds to a doublon configuration, and value zero for all other 2-particle initializations (see Sec. III C for definition of $C_{ij}$). We recall from Sec. III C that $\mu_2 = S_2 T/L^2$, where $S_2 \equiv \sum_{i<j} C_{ij}T/L^2$. Since there are $L^2/a^2$ distinct doublon configurations, where $L$ denotes the physical dimension of the lattice, we find that $S_2 = -2L^2/T$. Thus, $\mu_2 = -2$ in the limit $W = 0$, $V \to \infty$. From the discussion in Sec. III, we expect the quantization of $\mu_2$ to persist for finite disorder, $W$, and finite (but large) values of the interaction strength, $V$.

The discussion above shows that the model in Eq. (24) is characterized by two distinct values of the invariant $\mu_2$ in the limits where $V = 0$ and $V \to \infty$, respectively. Due to the robust quantization of $\mu_2$, which is protected by 2-particle localization, we hence conclude that the system supports two distinct topological phases that arise when $V \ll J$ and $V \gg J$, respectively. The transition between the phases is separated by a critical point, $V_{\mathrm{c}}$ [42]: when $V$ is increased past $V_{\mathrm{c}}$ in the thermodynamic limit, the

localization length in the two-particle sector should diverge at $V = V_c$, while $\mu_2$ changes abruptly from 0 to $-2$.

### A. Numerical simulations

Here we substantiate the discussion above through numerical simulations of the model: we first consider the limit of weak interactions, and show that the (quantized) average magnetic moment per particle remains unaffected by the nonzero interaction strength, as our analytical discussion predicts for an AFI phase with $\mu_1 = 2$. Subsequently, we show that that the model is characterized by a quantized, nonzero value of the invariant $\mu_2$, when $V$ is large, demonstrating that the system is in a CIAFI phase, distinct from the $\mu_1 = 2, \mu_2 = 0$ AFI phase.

#### 1. Weak interactions: AFI phase with $\mu_1 = 2$

We first present data from simulations of the model described above, in the limit of weak interactions. We consider a single disorder realization of the model with parameters $W = 2\pi/T$, $V = 0.1\,W$, and $\eta = 1/16$. From Ref. 37, we expect the model is many-body localized with these parameters. Since the model is obtained by adding weak interactions to a model of the AFAI with winding number 2 (see Refs. 30 and 31; here the factor of 2 arises because of the spin degeneracy), we moreover expect the system to be in the $\mu_1 = 2$ AFI phase (i.e., with $\mu_\ell = 0$ for $\ell > 1$).

To probe the topology of the system, we compute the mean magnetic moments of random time-evolved 4-particle states in a lattice of $6 \times 6$ sites. The long-time averaged magnetic moment, introduced in Sec. III, is defined as $\bar{M} = \sum_p a^2 \bar{m}_p$. The mean expectation value of $\bar{M}$, averaged over randomly chosen $\ell$-particle states (i.e., states chosen randomly from a given orthonormal basis) is given by $M_0[\ell] \equiv \binom{D}{\ell}^{-1}\mathrm{Tr}_\ell\,\bar{M}$, where the binomial coefficient $\binom{D}{\ell}$ counts the number of possible $\ell$-particle states in the system of $D = 2L^2$ single-particle states (here the factor of 2 arises due to the spin degeneracy, and $L = 6$ for the case we consider). Using that $\mathrm{Tr}_\ell\bar{M} = z_\ell L^2/T$, along with Eq. (18), we can express $M_0[\ell]$ in terms of the topological invariants $\mu_1 \ldots \mu_\ell$: $M_0[\ell] = \frac{L^2}{T}\sum_{\nu=1}^{\ell} A_\nu \mu_\nu$, where $A_\nu = \binom{D-\nu}{\ell-\nu}/\binom{D}{\ell}$. For $\ell = 4$ particles, our expectation that $\mu_1 = 2$ while $\mu_\ell = 0$ for $\ell > 1$ hence would lead to

$$M_0 = \frac{4a^2}{T}, \tag{27}$$

corresponding to an average magnetic moment per particle of $a^2/T$. This result was previously established for the noninteracting limit of the model (where the system is in the AFAI phase) [30]. The discussion above hence shows that the quantized average magnetic moment per particle in the AFAI is unaffected by interactions, as long as the system remains in the AFI phase.

To compute $M_0$ in the simulation, we pick as initial states 1972 random configurations of four particles located on individual sites. We evolve each initialization for 5,000 driving periods with a fixed disorder realization (the same for all initial states). Fig. 3b shows the particle density in the resulting final state for one of the realizations, after evolution for 5,000 periods. White dots and arrows indicate the corresponding initial configuration of occupied sites and spins. Note that the particle density remains non-uniform and confined near the initial location of the particles, consistent with many-body localization. We compute the time-averaged magnetic moment $\langle \bar{M} \rangle$ for each of the 1972 states, using the time-averaged bond-currents. The 1972 values of $\langle \bar{M} \rangle$ we obtained in this way are plotted in the histogram in Fig. 3c. Fig. 3d shows the time-averaged bond currents and magnetization density in the system for the same state used in Fig. 3b, used to calculate the magnetization. The distribution of $\langle \bar{M} \rangle$ obtained from these initializations was found to have mean $3.999997\,a^2/T$ and standard deviation $\delta M = 0.001 a^2/T$, resulting in a standard deviation of the mean at $\delta M/\sqrt{1972} \approx 0.00003 a^2/T$. This result is consistent with a $\mu_1 = 2$ AFI phase [see Eq. (27)].

#### 2. Strong interactions: CIAFI phase with $(\mu_1, \mu_2) = (2, -2)$

We now demonstrate that strong interactions drive the model into a CIAFI phase with $\mu_2 = -2$. These data were briefly discussed in Sec. I. Here we present them in further detail.

To show that large interaction strength drives the model into the CIAFI phase, we keep $W$ and $\eta$ fixed, but vary $V$. We moreover consider a single disorder realization with $18 \times 18$ sites. For each value of $V$ we consider, we obtained the time-evolution over 1000 driving periods for between 179 and 324 randomly chosen initializations where the two particles were located on particular sites and had distinct spins [59].

To establish the existence of a phase transition between the AFI and CIAFI phase, we considered the localization length in the system. We measured this using the inverse participation ratio of the density in the final state that resulted from each of the initializations we considered, $\mathcal{P} \equiv (\sum_{\mathbf{r}} |\rho_{\mathbf{r}}|^2)^{-1}$, where $\rho_{\mathbf{r}} = \sum_{s=\uparrow,\downarrow} \langle \hat{c}_{\mathbf{r},s}^\dagger \hat{c}_{\mathbf{r},s} \rangle$ denotes the particle density on site $\mathbf{r}$ in the final state. When each particle is localized on a particular site, $\mathcal{P}$ takes the value $1/4$ (in the case of a doublon configuration) or $1/2$. In contrast, $\mathcal{P} = L^2/4$ indicates full delocalization (corresponding to $\rho_{\mathbf{r}} = 2/L^2$ for all $\mathbf{r}$). More generally, $\mathcal{P}$ can effectively be seen as $1/4$ times the number of sites where the final state has support. This motivates us to define the effective localization length of the system, $\xi_{\mathrm{IPR}}$, as the average value of $\sqrt{4\mathcal{P}a^2}$ obtained from the initializations we probed.

In Fig. 1d, we plot the above localization length of

the system, $\xi_{\rm IPR}$, as a function of $V$. As is evident in the figure, the localization length remains small for small values of $V$. This indicates that the $\mu_1 = 2$ AFI phase at $V = 0$ remains stable for finite values of the interaction strength, as was also suggested by the results in Sec. IV A 1. In the range between $V = J$ and $V = 10J$, the localization length diverges, consistent with a phase transition. For $V \gtrsim 10J$, the localization length becomes small again, indicating the system has transitioned back into a stable phase. The localization length appears to remain small as $V$ goes to $\infty$; we hence expect this new phase to be the $\mu_2 = -2$ CIAFI phase.

To verify the existence of two distinct phases (namely the $\mu_1 = 2$ AFI and the $\mu_1, \mu_2 = 2, -2$ CIAFI phases), we computed the sum $S_2 \equiv \sum_{i<j} C_{ij}$, where $C_{ij} = \bar{M}_{ij} - \bar{M}_i - \bar{M}_j$ (see Sec. III C or I for definition of these quantities). In Fig. 1c, we plot the value of this sum. The data shows a clear transition between $\mu_2 = 0$ to $\mu_2 = -2$ in the range $V = J$ to $V = 10J$, where the localization length diverges. This further supports the existence of a $\mu_1, \mu_2 = 2, -2$ CIAFI phase for strong interactions, which is distinct from the AFI phase.

## V. DISCUSSION

In this work, we characterized the topological properties of periodically driven systems of interacting fermions in two dimensions. We established that the quantized magnetization of the AFAI persists in its interacting generalization, the anomalous Floquet insulator (AFIs). As a second result, we identified a new class of intrinsically-correlated nonequilibrium phases, namely the correlation-induced anomalous Floquet insulators (CIAFIs). The topological invariants characterizing the CIAFIs are encoded in the multi-particle correlations of the time-averaged magnetization density. While this work focused on driven fermionic models and their bulk topological invariants, our discussion can be readily extended to bosonic systems with particle number conservation.

Importantly, the topological protection of the CIAFIs does not require full many-body localization, but rather relies on *k-particle localization*, where the system is localized for any finite number of particles up to a maximum number, $k$. The existence of $k$-particle localization is well-established [42]. Since the existence of the CIAFI does not rely on full many-body localization, we may expect the behavior described above to be manifested via experimental signatures in the prethermal dynamics of systems which eventually thermalize at long times. Searching for other models that give rise to nontrivial values of these invariants and characterizing the physical properties that they imply will be interesting directions for future studies.

We demonstrated that CIAFIs may be realized in a tight-binding model with Hubbard type-interactions subject to a stepwise driving protocol. Recently, a noninter-

acting version of such a model was experimentally realized with ultracold atoms in optical lattices [60]. The CIAFI phases may be achieved in a similar experiment by adding Hubbard-type interactions to the system. We expect this type of interactions is natural to implement with ultracold atoms in optical lattices. Thus, we speculate that experimental realization of CIAFI phases is feasible with current experimental platforms.

At this point it is not clear whether the CIAFI phases are compatible with MBL, i.e., if they can exist in the thermodynamic limit of $L \to \infty$ and $k \to \infty$. (For finite $k$, localization is possible, and the physics described above is rigorously applicable.) In particular, we expect that CIAFI phases will exhibit dynamics strongly dependent on the initial state. In the model of Sec. IV, initial states where some large region $\mathcal{R}$ is doubly occupied would support chiral edge states moving around such regions. If the initial state contains such "internal edges," they may thermalize and serve as a weak heat bath for the remainder of the system. Next, if the density of filled regions $\mathcal{R}$ in the system is increased, we expect that at some point thermalizing internal edges will form a connected network, destroying localization. In contrast, initial states without filled, connected regions are expected to be much more stable, since there are no direct thermalization processes which involve few nearby particles; thermalization, if it occurs at all, will proceed either due to rare thermal inclusions, or due to multi-particle tunneling into, e.g., a state with "internal edges."

After the initial posting of this work, another preprint independently classified the bulk topological properties of two-dimensional MBL systems, when particle number conservation was present [61]. Interestingly the classification in Ref. 61 did not contain the CIAFI phases, suggesting that CIAFI phases and MBL may be incompatible. A definite answer for this question, however, remains lacking, and will be an interesting direction for future studies. In any case, the features above suggests that CIAFI phases (rigorously established for finite particle number) may provide a versatile playground for studying the interplay of weak thermalizing baths and MBL regions, which is expected to give new insights into the stability of MBL in 2d.

The topological classification we developed in the present work relied on particle number conservation. Chiral phases of spins and bosons without particle number conservation, which are close relatives of the AFAI (with higher-order invariants being zero, $\mu_\ell = 0$, $\ell > 2$), were considered in Ref. 29. It was shown that, when many-body localized, such phases are characterized by a quantized topological index which describes the pumping of quantum information along the edge over one driving period. Such an index arises from the rigorous classification of anomalous local unitary operators in one-dimensional systems, developed by Gross et al [62]. It will be an interesting direction of future studies to investigate whether the bulk classification of the present work can be generalized to systems where particle conservation

is not present.

In the future, it will moreover be interesting to investigate how thermalization is manifested in experimentally realistic situations for the CIAFI phases, and what the corresponding time scales are. With $k$-particle localization present (for some large $k$), thermalization must be driven by correlated processes involving more than $k$ particles. It is natural to expect that such thermalizing process will be parametrically slow, and therefore signatures of the CIAFI phases (and the AFI), such as quantization of magnetization, would be observable even if MBL is eventually destroyed. A systematic study of such thermalization timescales will be an interesting question for future studies, with significance beyond the context of topological phases we considered here.

*Acknowledgements* — M.R. and F.N. thank the Villum Foundation and the European Research Council (ERC) under the European Union Horizon 2020 Research and Innovation Programme (Grant Agreement No. 678862) for support. D.A. acknowledges support by the Swiss National Science Foundation. N.L. acknowledges support from the European Research Council (ERC) under the European Union Horizon 2020 Research and Innovation Programme (Grant Agreement No. 639172), and from the Israeli Center of Research Excellence (I-CORE) "Circle of Light". M.R. and E.B. acknowledge support from CRC 183 of the Deutsche Forschungsgemeinschaft.

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

alizations where resonances occur between two or more sites separated by a distance comparable to the system size, $L$. Disorder realizations supporting such accidental resonances do not meet the conditions for $k$-particle or many-body localization, as defined in Sec. II. However, for a randomly chosen disorder realization within the $k$-particle localized region of parameter space, the probability that the $\ell$-particle quasienergy spectrum (for each $\ell \leq k$) features any such an accidental resonance goes to zero in the thermodynamic limit $L \to \infty$ [42]. In the following, we assume that the disorder realization under consideration does not feature such accidental resonances; within the $k$-particle localized regime of parameter space, this assumption holds with probability 1 in the thermodynamic limit.

[44] As for the $k = 1$ (i.e., single-particle) special case [30, 31], we expect that $k$-particle localization in the bulk can coexist with delocalized edge states [42]. A detailed study of the interplay between bulk localization and delocalized edge states in the case of full MBL is left for future work; some aspects have been discussed in Refs. 37 and 63.

[45] M. Serbyn, Z. Papic, and D. A. Abanin, Physical Review Letters **111** (2013).

[46] D. A. Huse, R. Nandkishore, and V. Oganesyan, Phys. Rev. B **90**, 174202 (2014).

[47] Note that, due to the finite Lieb-Robinson velocity of the system, the conservation of the LIOMs by the stroboscopic evolution requires the system's dynamics to also be localized at intermediate times.

[48] D. A. Abanin, E. Altman, I. Bloch, and M. Serbyn, Rev. Mod. Phys. **91**, 021001 (2019).

[49] Note that, with this notation, each bond $b$ is implicitly defined with an orientation, such that $I_b(t)$ measures the current along the bond's orientation.

[50] This follows from straightforward generalization of the arguments in Ref. 56 to periodically driven systems. In particular, note that our assumption of localization by definition precludes the possiblity of resonances between far-separated sites that cause $\bar{I}_b$ to have support far away from the bond $b$ (see also Footnote 43).

[51] Note that Ampere's law is only meaningful when the current density has zero divergence. The long-time-averaged magnetization density $\bar{m}_p$ in an MBL system is always well-defined, since the time-averaged current density always has zero divergence. Moreover, while Ampere's law only defines magnetization density up to a constant shift, $\bar{m}_p$ is uniquely defined by the definition in Sec. III A.

[52] For a finite system, the fact that $m^p_{\alpha_1 \dots \alpha_\ell}$ is exponentially insensitive to the details of the system far away from the plaquette $p$ means that it may only change by an amount of order $e^{-L/\xi}$ when the system size is increased. This implies that the sum $\Sigma_{\alpha_1 \dots \alpha_\ell} m^p_{\alpha_1 \dots \alpha_\ell}$ is given by its value in the thermodynamic limit, up to a correction of order $e^{-L/\xi}$.

[53] Here the region of support is understood as the region of the lattice where the particle density is significant in the state $|\psi_n\rangle$. See Appendix B for further details.

[54] For noninteracting systems, $\bar{m}_p$ is a one-body operator, and $m^p_{\alpha_1 \dots \alpha_k} = 0$ when $k \geq 2$. Hence only $\mu_1$ may be nonzero [see Eq. (21)]. In this case, Eq. (18) implies that $\mathrm{Tr}_k \bar{m}_p = \binom{D-1}{k-1} \mu_1$. Using $D = L^2$, we see that $\binom{D-1}{k-1}$ scales as $L^{2(k-1)}$ with the system size $L$.

[55] In case of degeneracies, one can always pick a basis of eigenstates where $\bar{m}_p$ is diagonal.

[56] A. Chandran, I. H. Kim, G. Vidal, and D. A. Abanin, Physical Review B **91** (2015).

[57] To see this, note that $\hat{n}_\alpha |\Psi_\mathcal{R}\rangle = (1 - f_\alpha f^\dagger_\alpha)|\Psi_\mathcal{R}\rangle$. The operator $f^\dagger_\alpha$ is a polynomial in $\{c_\alpha\}$ and $\{c^\dagger_\alpha\}$, where each term has the net effect of creating one fermion in the region around LIOM $a$. Since all sites near the LIOM $a$ are occupied for the state $|\Psi_\mathcal{R}\rangle$, $f^\dagger_\alpha |\Psi_\mathcal{R}\rangle = 0$, and thus $\hat{n}_\alpha |\Psi_\mathcal{R}\rangle = |\Psi_\mathcal{R}\rangle$.

[58] To see that the sum in Eq. (23) converges, note that the coefficient $m_{\alpha_1 \dots \alpha_\ell}$ is exponentially suppressed in $d/\xi$, where $d$ is the distance from any of the LIOM centers $\mathbf{r}_{\alpha_1} \dots \mathbf{r}_{\alpha_\ell}$ to the plaquette $p$. The number of distinct LIOMs whose centers are located within a radius $\xi_l$ from the plaquette $p$ is of order $\xi_l^2/a^2$, where $a$ is the lattice constant in the system. Therefore, the coefficient $m^p_{\alpha_1 \dots \alpha_\ell}$ vanishes exponentially when $\ell \gg \xi_l^2/a^2$. Recalling that $\mu_\ell \equiv \Sigma_{\alpha_1 \dots \alpha_\ell} m^p_{\alpha_1 \dots \alpha_\ell}$ must take integer value for each $\ell$, we thus conclude that $\mu_\ell$ equals zero when $\ell \gg \xi_l^2/a^2$.

[59] More precisely, the initializations were divided into three classes, that contained initializations where the two particles were located on the same site, adjacent sites, and all other "on-site" initializations, respectively. The average magnetization was found through the sum of the obtained mean values of $\langle \bar{M} \rangle$ within each class, weighted according to the number of states in the class.

[60] K. Wintersperger, C. Braun, F. N. Ünal, A. Eckardt, M. D. Liberto, N. Goldman, I. Bloch, and M. Aidelsburger, Nat. Phys. **16**, 1058 (2020), ISSN 1745-2481.

[61] C. Zhang and M. Levin, arXiv (2020), 2010.02253.

[62] D. Gross, V. Nesme, H. Vogts, and R. F. Werner, Communications in Mathematical Physics **310**, 419 (2012).

[63] R. Nandkishore and S. Gopalakrishnan, Annalen der Physik **529**, 1600181 (2016).

[64] Specifically, we require that $\lim_{L \to \infty} L^{p'} \mathcal{O}(L^{-p}) = 0$ for $p' < p$.

[65] The fact that the total flux on the torus is given by an integer multiple of $2\pi$ means that this requirement does not require a specification of the interior region of the loop.

[66] Specifically, the spectral norm of an operator $M$ is defined as $\|M\| \equiv \sup_{|\psi\rangle} \sqrt{\langle \psi | M^\dagger M | \psi \rangle / \langle \psi | \psi \rangle}$.

[67] M. L. Mehta, *Random Matrices*. (Elsevier, Amsterdam, 2004).

[68] Here the tensor product $|\psi_1\rangle_A \otimes |\psi_2\rangle_B$ is defined as $\hat{C}^\dagger_1 \hat{C}^\dagger_2 |0\rangle$, where $C^\dagger_1$ is the unique combination of fermionic creation operators that creates the state $|\psi_1\rangle_A$, i.e., $|\psi_1\rangle_A = C^\dagger_1 |0\rangle_A$, where $|0\rangle_A$ denotes the vacuum in subsystem $A$. $C^\dagger_2$ is defined in a similar fashion.

## Appendix A: Proof of Eq. (4)

In this appendix we establish that the time-averaged current that passes through a cut $C$ between two plaquettes $p$ and $q$ is determined by two quasilocal operators, $\bar{m}_p$ and $\bar{m}_q$, with support centered at $p$ and $q$, respectively [see Eq. (4) and Fig. 4]. By considering two plaquettes separated by a distance much longer than the localization length, this provides a prescription for uniquely identifying the magnetization density operator $\bar{m}_p$ (up

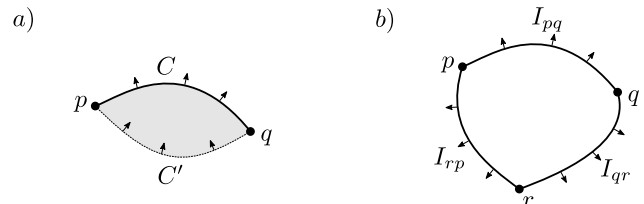

FIG. 4. a) Schematic depiction of the argument showing that time-averaged current through a cut $C$ between to plaquettes $p$ and $q$ only depends on the cut's two end-points. Specifically, since there can be no accumulation of charge over time in the region between the cuts $C$ and $C'$, the same current must pass through the two cuts, and thus $\bar{I}_C = \bar{I}_{C'}$ for any two cuts $C$ and $C'$ between the plaquettes $p$ and $q$. b) The vanishing divergence of current implies that $\bar{I}_{C_{pq}} + \bar{I}_{C_{qr}} = \bar{I}_{C_{pr}}$.

to exponentially small corrections in the distance, which can be of order the system size).

We recall from the main text that the operator corresponding to current through the cut $C$ is given by

$$I_C(t) = \sum_{b \in B_C} I_b(t), \qquad (A1)$$

where $I_b$ denotes the bond current operator on bond $b$, and the sum runs over all bonds that cross the cut $C$.

The goal of this Appendix is to find the time-averaged expectation value of the current, $\langle\langle I_C \rangle\rangle$, resulting from some given initial state $|\psi\rangle$. As in the main text, we use $\langle\langle \mathcal{O} \rangle\rangle \equiv \lim_{\tau \to \infty} \frac{1}{\tau} \int_0^\tau dt\, \langle \psi(t)|\mathcal{O}(t)|\psi(t)\rangle$. The time-averaged expectation value of the current $I_C$ may equivalently be computed in the Heisenberg picture as $\langle\langle I_C \rangle\rangle = \langle \psi|\bar{I}_C|\psi\rangle$, where $|\psi\rangle$ denotes the initial state of the system. Here, as in the main text, for any Schrödinger picture operator $\mathcal{O}(t)$ [such as $I_C(t)$], $\bar{\mathcal{O}}$ denotes the time-average of the current $I_C$ in the Heisenberg picture,

$$\bar{\mathcal{O}} \equiv \lim_{\tau \to \infty} \frac{1}{\tau} \int_0^\tau dt\, U^\dagger(t)\mathcal{O}(t)U(t). \qquad (A2)$$

The time-averaged current operator $\bar{I}_C$ is thus obtained by transforming the time-dependent operator $I_C(t)$ in Eq. (A1) with evolution operator $U(t)$, and integrating over time as in Eq. (A2).

To explore the properties of $\bar{I}_C$, we consider the time-averaged current for a different cut, $C'$, between the same two plaquettes $p$ and $q$, see Fig. 4a. We note that $I_C(t) - I_{C'}(t) = \dot{N}_R(t)$, where $N_R$ measures the number of particles in the region $R$ between cut $C$ and $C'$ (shaded region in Fig. 4). Importantly, since $N_R$ is bounded by the number of sites in the region $R$, the long-time-averaged value of $\langle \dot{N}_R \rangle$ must vanish. We thus conclude that $\langle\langle I_C \rangle\rangle = \langle\langle I_{C'} \rangle\rangle$. Since this holds for any initial state $|\psi\rangle$, we conclude that

$$\bar{I}_C = \bar{I}_{C'}. \qquad (A3)$$

As a next step, we note from Eqs. (A1) that $\bar{I}_C = \sum_{b \in B_C} \bar{I}_b$, where $\bar{I}_b$ denotes the time-averaged current

on bond $b$ [see Eq. (A2)]. We note that the operator $I_b(t)$ is local, with support only on the sites connected by the bond $b$. For many-body localized systems, this implies that the operator $\bar{I}_b$ is a localized integral of motion, with support within a distance $\sim \xi_l$ from the bond $b$, up to an exponentially small correction [50]. Hence, $\bar{I}_C$ is given by a sum of terms, each of which only has support within a region of radius $\xi_l$, centered at a point along the cut $C$.

The requirements that $\bar{I}_C$ is given by a sum of local terms as described above, while at the same time taking the same value for all cuts between plaquettes $p$ and $q$ [Eq. (A3)], significantly constrains the form that $\bar{I}_C$ can take. In particular, this implies that $\bar{I}_C = I(p, q)$, where the operator $I(p, q)$ only depends on the locations of the two plaquettes $p$ and $q$ (and not on the details of the cut $C$). Moreover, for any cut between plaquettes $p$ and $q$, $I(p, q)$ is given by a sum of terms which only have support in a region of width $\xi_l$ around the cut. For any site located a distance larger than $\xi_l$ from both plaquettes $p$ or $q$, we can find a cut that remains separated from the site by a distance larger than $\xi_l$. Therefore the support of operator $I(p, q)$ can only include sites within a localization length of the endpoints $p$ and $q$. Hence, we write:

$$I(p, q) = A_1(p, q) + A_2(p, q), \qquad (A4)$$

where $A_1(p, q)$ has its full support within a region of width $\xi_l$ around plaquette $p$, and $A_2(p, q)$ has support around plaquette $q$. The operators $A_1(p, q)$ and $A_2(p, q)$ depend only on the locations of plaquettes $p$ and $q$, respectively.

By letting the cut from $p$ to $q$ go through an arbitrary plaquette $r$ on the torus (see Fig. 2b), we conclude from the arguments above the $I(p, r) + I(r, q) = I(p, q)$. This implies

$$A_1(p, r) + A_2(p, r) + A_1(r, q) + A_2(r, q) = A_1(p, q) + A_2(p, q). \qquad (A5)$$

The only terms on the left hand side with support near plaquette $r$ are the terms $A_2(p, r)$, and $A_2(r, q)$, while none of the terms on the right-hand side have support near plaquette $r$. We thus conclude that $A_2(p, r) = -A_1(r, q)$ for any choice of two plaquettes $p$ and $q$. Hence we may write $A_1(r, q) = A(r)$, and $A_2(p, r) = -A(r)$ for some function $A(r)$ which only depends on the location of plaquette $r$ and has its full support near plaquette $r$. Using this in Eq. (A4), we find

$$I(p, q) = A(p) - A(q). \qquad (A6)$$

Identifying $A(p) = \bar{m}_p$, we thus conclude that Eq. (4) holds.

## Appendix B: Derivation of Eq. (10)

Here we derive Eq. (10), which is used to establish the integer quantization of the topological invariant $z_\ell$.

To recapitulate, we consider a $k$-particle localized system, where $k$ may be infinite in the case of full MBL. For a given $\ell \leq k$, we consider the $\ell$-particle Floquet eigenstates of the system, $\{|\psi_n\rangle\}$, with corresponding quasienergies $\{\varepsilon_n\}$, and let $\tilde{\varepsilon}_n$ denote the perturbed quasienergy corresponding to $\varepsilon_n$ when the weak uniform magnetic field $B_0 = 2\pi/L^2$ is inserted that results in one flux quantum piercing the torus (see below for details). The goal of this Appendix is to establish two results. First, we show that for each $\ell$-particle Floquet eigenstate, $|\psi_n\rangle$,

$$\tilde{\varepsilon}_n = \varepsilon_n - B_0\langle\psi_n|\bar{M}|\psi_n\rangle + \mathcal{O}(L^{-5/2}). \quad (B1)$$

Here, and in the remainder of this Appendix, $\mathcal{O}(L^{-p})$ indicates a correction which goes to zero at least as fast as $L^{-p}$ [64]. (I.e, in the following, we only indicate how rapidly corrections decrease with system size.) Secondly, we show that, when summed over all $\ell$-particle Floquet states, the corrections of order $L^{-5/2}$ in Eq. (B1) approximately cancel out, yielding a net correction which is *exponentially* suppressed in system size:

$$\sum_n (\tilde{\varepsilon}_n - \varepsilon_n) = -\sum_n B_0\langle\psi_n|\bar{M}|\psi_n\rangle + \mathcal{O}(e^{-L/\xi}), \quad (B2)$$

where $\mathcal{O}(e^{-L/\xi})$ likewise indicates a correction that goes to zero as $e^{-L/\xi}$ in the thermodynamic limit.

Eqs. (B1) and (B2) implicitly require that, for each quasienergy level $\varepsilon_n$ of the (unperturbed) zero-flux system, it should be possible to identify a unique quasienergy level $\tilde{\varepsilon}_n$ of the (perturbed) one-flux system which satisfies Eq. (B1). In Sec. B 4 below, we confirm that such a complete one-to-one identification is possible for all but a set of disorder realizations which has measure zero in the thermodynamic limit.

As noted in the main text, Eq. (B2) does not follow trivially from first-order perturbation theory in the weak magnetic field $B_0$: under a continuous perturbation of the system, the system's quasienergy spectrum undergoes exponentially many avoided crossings due to resonances between many-body Floquet eigenstates separated by a large distance in Fock space. Hence, first-order perturbation theory breaks down for the system. Instead, we establish Eq. (10) with an alternative approach, using the localization properties of the many-particle Floquet eigenstates.

In order to follow this approach, we use a succession of auxiliary results which are not discussed in detail in the main text, but are crucial for the proof of Eqs. (B1) and (B2). The line of arguments proceeds as follows: we first show explicitly how the uniform magnetic field $B_0$ can be implemented in the system (Sec. B 1). Subsequently, in Sec. B 2 we show that, for a given finite region $S$ of the lattice, it is always possible to choose a gauge where the Hamiltonian $\tilde{H}$ of the one-flux system resembles the Hamiltonian $H$ of the zero-flux system locally within $S$, and likewise for the Floquet operators $\tilde{U}$ and $U$ (Sec. B 3). Using this result, we

demonstrate in Sec. B 4 that the Floquet eigenstates and quasienergies, $\{|\psi_n\rangle\}$ and $\{\varepsilon_n\}$, are robust to the perturbation caused by inserting of the weak uniform magnetic field $B_0$, such that the one-to-one identification described above is possible. From these auxilliary results, we prove Eq. (B1) in Sec. B 5, and finally use Eq. (B1) along with the LIOM structure of the system to establish Eq. (B2) (Sec. B 6).

For the sake of brevity, throughout this Appendix we will work with a fixed degree of localization and particle number, unless otherwise noted. Thus, in the following, $k$ and $\ell$ are fixed constants that refer to the system's degree of localization and to the number of particles in the system, respectively. We take $\ell \leq k$ in the discussion below.

### 1. Implementation of magnetic flux

Here we discuss how the magnetic flux is implemented. The system we consider consists of interacting fermions on a lattice with the geometry of a torus, of dimensions $L \times L$. The Hamiltonian of the system (in the absence of a flux) takes the form

$$H(t) = \sum_{ij} J_{ij}(t)\hat{c}_i^\dagger \hat{c}_j + H_{\text{int}}(t), \quad (B3)$$

where $\hat{c}_i$ annihilates a fermion on site $i$ in the lattice. Here the first term contains both hopping and on-site potentials, including disorder, with $J_{ij}(t) = J_{ji}^*(t)$, while the term $H_{\text{int}}$ accounts for interactions. We allow both parts of the Hamiltonian to be time-dependent, with periodicity $T$. To simplify the discussion, we consider the case of a square lattice model with nearest-neighbour hoppings, and a density-density interaction described by $H_{\text{int}} = \frac{1}{2}\sum_{i,j} \hat{\rho}_i\hat{\rho}_j V_{ij}(t)$, where $\hat{\rho}_i = \hat{c}_i^\dagger \hat{c}_j$ and $V_{ij}(t) = V_{ji}(t)$ is real. In the general case of a quasilocal Hamiltonian, the results below can also be derived using similar arguments.

In this subsection we are interested in finding the Hamiltonian $\tilde{H}(t)$ of the system when the uniform magnetic field $B_0 = \frac{2\pi}{L^2}$ is inserted, corresponding to one flux quantum through the surface of torus. Having assumed $H_{\text{int}}(t)$ to consist of density-density interactions, only the first term in Eq. (B3) is affected by the magnetic flux. The Hamiltonian $\tilde{H}(t)$ thus takes the form:

$$\tilde{H}(t) = \sum_{ij} e^{i\theta_{ij}} J_{ij}(t)\hat{c}_i^\dagger \hat{c}_j + H_{\text{int}}(t). \quad (B4)$$

Here, the Peierls phases $\{\theta_{ij}\}$, with $\theta_{ij} = -\theta_{ji}$, must ensure that the total phase acquired by traversing a closed loop on the torus is given by $B_0 A_S \pmod{2\pi}$, where $A_S$ is the area enclosed by the loop [65].

There are (infinitely) many distinct configurations of the phases $\{\theta_{ij}\}$ that satisfy this condition, corresponding to different choices of gauge for the one-flux Hamiltonian $\tilde{H}(t)$. As the starting point for the following discussion, we consider the following Landau-type gauge: let $\theta_i^x$

denote the Peierls phase for hopping along the bond in the positive $x$-direction from site $i$ (and similarly let $\theta_i^y$ be the Peierls phase for hopping in the positive $y$-direction), and give them the values:

$$\theta_i^y = B_0 x_i a \quad \theta_i^x = B_0 L y_i \delta_{x_i, L}. \tag{B5}$$

Here $x_i$ and $y_i$ denote the coordinates of site $i$ (defined with branch cut outside $S_0$), and $\delta_{ij}$ denotes the Kronecker delta symbol, such that $\delta_{x_i, L}$ takes the value 1 if $x_i = L$, while $\delta_{x_i, L} = 0$ for all other values of $x_i$. Recall that $a$ is the lattice constant. The phases $\theta_i^y$ ensure that a trajectory encircling a plaquette acquires a phase of $B_0 a^2$, if the trajectory does not cross the branch cut of the $x$-position operator between $x = L$ and $x = 0$. The phase $\theta_i^x$, which does not appear in the Landau gauge in an open geometry, is necessary to ensure that the phase is also given by $B_0 a^2 \pmod{2\pi}$ for trajectories encircling plaquettes across the branch cut.

The goal of the following is to show that we can choose another gauge where $B_0$ only weakly perturbs the Hamiltonian within a particular finite region of the lattice, $S$, which consists of one or more non-overlapping disk-shaped regions, $S_1, \ldots S_N$, whose combined area, $A_S$, is much smaller than $L^2$. We reach such a gauge through the following transformation to the one-flux Hamiltonian with the gauge choice as prescribed in Eq. (B4): $\hat{c}_i \to e^{-i\phi_i} \hat{c}_i$, where $\phi_i = B_0 x_0^{(n)} y_i$ for sites $i$ within subregion $S_n$, and $(x_0^{(n)}, y_0^{(n)})$ denotes the center of subregion $S_n$. In this case, one can verify that, for sites within subregion $n$ the Peierls phases resulting from this transformation take the following values:,

$$\theta_i^y = B_0(x_i - x_0^{(n)}), \quad \theta_i^x = 0. \tag{B6}$$

The later holds since the branch cut of the $x$-coordinate does not intersect $S$. Since $S_n$ has disk geometry and is centered around $(x_0^{(n)}, y_0^{(n)})$, we thus find $|x_i - x_0^{(n)}| \leq \sqrt{A_S}$ for sites $i$ within subregion $S_n$. Hence we confirm that the Peierls phases are all of order $\sqrt{A_S} a/L^2$ for bonds within $S$, and thus much smaller than 1 in the limit $A_S \ll L^2$ specified above.

### 2. Response of the Hamiltonian

An important result we will use extensively in the following is that, for large systems, the insertion of the uniform field $B_0$ only weakly perturbs the system, up to a gauge transformation. To see this, we consider the action of the perturbation induced by $B_0$, $\delta H(t) \equiv \tilde{H}(t) - H(t)$ (in the particular gauge we consider), on a state $|\psi\rangle$ with an arbitrary number of particles, where all particles are located in the finite region $S$ that was introduced in the previous subsection.

As a first step, we note that $\delta H(t)|\psi\rangle = \delta H(t) P_S |\psi\rangle$, where $P_S$ projects into the subspace where all particles are located within $S$. Using that $\hat{c}_i P_S = 0$ if site $i$ is

located outside $S$, we find

$$\delta H(t) P_S = \sum_{j \in S} \sum_i J_{ij}(t) \hat{c}_i^\dagger \hat{c}_j (e^{i\theta_{ij}} - 1). \tag{B7}$$

The Peierls phases $\{\theta_{ij}\}$ are as given in Eq. (B5) above. Below, we establish an upper bound for the spectral norm [66] of $\delta H(t) P_S$, $\|\delta H(t) P_S\|$. To do this, we use that $\|M\| \leq \sqrt{\text{Tr}(M^\dagger M)}$, such that

$$\|\delta H P_S\|^2 \leq \sum_{j_1, j_2 \in S} \sum_{i_1, i_2} K_{i_1 j_1}^* K_{i_2 j_2} \text{Tr}(\hat{c}_{j_1}^\dagger \hat{c}_{i_1} \hat{c}_{i_2}^\dagger \hat{c}_{j_2}),$$

where $K_{ij} \equiv J_{ij}(e^{i\theta_{ij}} - 1)$, and we suppressed time-dependence for brevity. Since $\theta_{ij} = 0$ for $i = j$, terms above are only nonzero when $i_1 = i_2$ and $j_1 = j_2$. Thus,

$$\|\delta H P_S\|^2 \leq \sum_{j \in R} \sum_i |J_{ij}|^2 |e^{i\theta_{ij}} - 1|^2. \tag{B8}$$

We now estimate the maximal scale of the right hand side above. We recall from the discussion in the end of Subsection B 1 that the Peierls phases $\{\theta_{ij}\}$, as given in Eq. (B5), are of order $\sqrt{A_S} a/L^2$ or smaller for bonds within the region $S$. This implies that the value of each non-vanishing term in the sum in Eq. (B8) is of order $J^2 A_S a^2/L^4$ or less, where $J$ denotes the typical scale of the (off-diagonal) tunneling coefficients $\{J_{ij}\}$. To estimate the number of non-vanishing terms in the sum we recall, from the assumptions made in the beginning of subsection B 1, that the tunneling coefficients $J_{ij}$ only couple nearest-neighbor pairs of sites in the lattice. Hence, for each choice of the index $j$, $J_{ij}$ may only be non-vanishing for four choices of the index $j$. These considerations show that there are only of order $A_R/a^2$ non-vanishing terms in the sum above. Using that each non-vanishing term has norm of order $\lesssim J^2 A_R a^2/L^4$, we find that $\|\delta H P_S\|^2 \lesssim A_S^2 J^2 L^{-4}$. Here $a \lesssim b$ indicates that $a$ is smaller than $b$, or of order $b$. Thus we conclude that

$$\|\delta H P_S\| \lesssim J A_S/L^2. \tag{B9}$$

In the sense of the operator norm, the difference between the Hamiltonians with and without one flux quantum uniformly piercing the entire torus decays to zero with the inverse of the total system area, when acting on states confined to the region $S$, and with a judicious choice of gauge.

#### a. Action on a localized state

Using the above result, we now show that a gauge exists where $\delta H$ is small when acting on states which are not strictly confined to the region $S$ of the lattice, but rather only exponentially localized. Specifically, we consider a state $|\psi\rangle$, whose full support is exponentially confined to a region $S$ which consists of one or more disk-shaped subregions of radius $r$, with the probability of

finding a particle a distance $s$ from the center of the nearest subregion decaying as $e^{-s/\xi_l}$ when $s > r$.

To conveniently quantify the extent to which particles are confined within a subregion of the lattice, for each $j = 1, 2 \ldots$, we let $|\psi_j\rangle$ denote the component of the wavefunction $|\psi\rangle$ where the outermost particle is located in the distance interval between $(j-1)a$ and $ja$ from the nearest subregion of $S$. Specifically, $|\psi_j\rangle \equiv (P_j - P_{j-1})|\psi\rangle$, where $P_j$ denotes the projector onto the states where *all* particles are located within a distance $ja$ from the center of the nearest subregion of $S$. From this definition one can verify that $|\psi\rangle = \sum_{j=1}^{\infty} |\psi_j\rangle$. Moreover, the using that $P_j P_k = P_{\min(j,k)}$, it follows that the components are mutually orthogonal: $\langle \psi_j | \psi_k \rangle = 0$ for $j \neq k$. From the definitions above, the probability for finding finding a particle more than a distance $ja$ from the center of $R$ is given by $\langle \psi | (1 - P_j) | \psi \rangle = \sum_{j'=j+1}^{\infty} \langle \psi_{j'} | \psi_{j'} \rangle$. Since the left hand side must be of order $e^{-ja/\xi}$ for $ja > r$, and each term in the right hand side is positive, we must have

$$\langle \psi_j | \psi_j \rangle \lesssim e^{-ja/\xi_l} \quad \text{for} \quad j > r/a. \tag{B10}$$

We now use the above result to obtain a bound for the state $\delta H |\psi\rangle$. Inserting $|\psi\rangle = \sum_{j=1}^{\infty} |\psi_j\rangle$, and using $P_j |\psi_j\rangle = |\psi_j\rangle$ one can verify that $|\psi\rangle = P_R|\psi\rangle + \sum_{j>r/a} P_j |\psi_j\rangle$, where $P_S \equiv P_{r/a}$ denotes the projector into the subspace where all particles are located within the region $S$ (for convenience we assume $r$ to be an integer multiple of the lattice constant $a$). Using this result along with the triangle inequality and Eq. (B10), we hence obtain:

$$\|\delta H |\psi\rangle\| \lesssim \|\delta H P_R\| + \sum_{j>r/a} \|\delta H P_j\| e^{-\frac{ja}{2\xi_l}}.$$

The considerations from Sec. B 2 show that we may choose a gauge for $\tilde{H}$ such that $\|\delta H P_S\| \lesssim J A_S/L^2$, and $\|\delta H P_j\| \lesssim A_{S_j}^2 J/L^2$ for any choice of $j$, where $A_{S_j} \sim (ja)^2$ denotes the area of the region projected into by $P_j$. Using that $\sum_{j>j_0} j^2 e^{-j/k} \sim j_0^2 e^{-j_0/k}$ when $j_0 \gg k$, one can then verify that

$$\sum_{j>r/a} \|\delta H P_j\| e^{-\frac{ja}{2\xi_l}} \lesssim A_S J/L^2 e^{-r/2\xi_l}, \tag{B11}$$

where $A_S \sim r^2$ denotes the area of the region $S$. Thus, since $r \gg \xi_l$, we find

$$\|\delta H |\psi\rangle\| \lesssim J A_S/L^2. \tag{B12}$$

### 3. Response of the Floquet operator

We now show that, for any region $S$ in the lattice that consists of one or more disk-shaped subregions, it is possible to find a gauge, the Floquet operators of the one- and zero-flux systems, $\tilde{U}(T)$ and $U(T)$, have nearly identical actions states $|\psi\rangle$ localized within $S$:

$\tilde{U}(T)|\psi\rangle \approx U(T)|\psi\rangle$. Here the state is said to be localized within $S$ if the probability of finding a particle a distance $s$ from the center of the nearest subregion os $S$ decays as $e^{-s/\xi_l}$ for $s > r$, where $r$ denotes the radius of $S$.

First, we note that $\|(U - \tilde{U})|\psi\rangle\| = \|(\tilde{U}^\dagger U - 1)|\psi\rangle\|$. This follows from the unitarity of $\tilde{U}$, since for any state $|\Psi\rangle$, $\||\Psi\rangle\| = \|\tilde{U}^\dagger |\Psi\rangle\|$. Using that $\tilde{U}^\dagger U - 1 = \int_0^T dt\, \partial_t [\tilde{U}^\dagger(t) U(t)]$, along with $\delta H(t) \equiv \tilde{H}(t) - H(t)$, we find

$$(U - \tilde{U})|\psi\rangle = -i \int_0^T dt\, \tilde{U}^\dagger(t) \delta H(t) U(t) |\psi\rangle. \tag{B13}$$

Using that $\||\Psi\rangle\| = \|\tilde{U}^\dagger |\Psi\rangle\|$ along with the triangle inequality, we thus find

$$\|(U - \tilde{U})|\psi\rangle\| \leq \int_0^T dt\, \|\delta H(t) U(t) |\psi\rangle\|. \tag{B14}$$

We now use that $U(t)$ is local at all times $0 \leq t \leq T$, due to the finite Lieb-Robinson velocity $v$ of the system. The locality implies that, for the state $U(t)|\psi\rangle$, the probability of finding a particle a distance $s$ from the center of $S$ decays exponentially when $s \gtrsim r$. Using the result in Eq. (B12) from the previous subsection, we thus find

$$\|\delta H(t) U(t) |\psi\rangle\| \lesssim J A_S/L^2. \tag{B15}$$

Using this in the inequality in Eq. (B14), we conclude

$$\|(U^\dagger \tilde{U} - 1)|\psi\rangle\| \lesssim J T A_S/L^2. \tag{B16}$$

Thus, $\|(\tilde{U} - U)|\psi\rangle\| \lesssim J T A_S/L^2$.

The result in Eq. (B16) shows that, with a judicious choice of gauge, the Floquet operators of the one- and zero flux systems give nearly identical results when acting on a localized state. In this sense, the insertion of a uniform magnetic field $B_0$ only weakly modifies the Floquet operator for large systems.

### 4. Response of Floquet eigenstates and quasienergy spectrum

We now show that, in the subspace with $k$ or fewer particles, the quasienergy spectrum and Floquet eigenstates of $k$-particle localized systems are robust to perturbations, and only weakly affected by the insertion of the uniform magnetic field $B_0$.

In this subsection, it is useful to use notation that relates the quasienergies and Floquet eigenstates to the LIOM decomposition in Eq. (1) (which is valid in the subspace of up to $k$ particles, which we consider): in the following we thus let $|\Psi_{\alpha_1 \ldots \alpha_\ell}\rangle \equiv \hat{f}_{\alpha_1}^\dagger \ldots \hat{f}_{\alpha_\ell}^\dagger |0\rangle$ denote the Floquet eigenstate of the system for which only LIOMs $\alpha_1 \ldots \alpha_\ell$ take value 1 (see Sec. II A for definition of $\hat{f}_\alpha^\dagger$), and let $E_{\alpha_1 \ldots \alpha_\ell}$ denote the corresponding quasienergy.

Using this cutoff length, we show below that for each finite $\ell \leq k$, where $k$ denotes the system's degree of localization (which is infinite for MBL systems), the $\ell$-particle Floquet eigenstates $\{|\tilde{\Psi}_{\alpha_1 \ldots \alpha_\ell}\rangle\}$ of $\tilde{U}$ can be labeled such that, for *each* choice of LIOMs (identified by the LIOM indices $\alpha_1 \ldots \alpha_\ell$),

$$|\tilde{\Psi}_{\alpha_1 \ldots \alpha_\ell}\rangle = |\Psi_{\alpha_1 \ldots \alpha_\ell}\rangle + \mathcal{O}\left(L^{-1/2}\right), \qquad \text{(B17)}$$

and

$$\tilde{E}_{\alpha_1 \ldots \alpha_\ell} = E_{\alpha_1 \ldots \alpha_\ell} + \mathcal{O}\left(L^{-2}\right). \qquad \text{(B18)}$$

Eq. (B17) thus shows that, in the thermodynamic limit, *each* eigenstate of $\tilde{U}$ is identical to an eigenstate of $U$, up to gauge transformation and a vanishingly small correction, while Eq. (B18) shows that their associated quasienergies similarly are identical up to a vanishing correction. This establishes the one-to-one correspondence of the quasienergy levels of the zero- and one-flux systems that we summarized below Eq. (B2).

Due to the possibility that the field $B_0$ induces a resonance between two Floquet eigenstates of $U$, disorder realizations do exist where one (or more) of the eigenstates of $\tilde{U}$ is a significantly hybridized combination of two eigenstates of $U$. In this case, Eq. (B17) will hold for most but not all Floquet eigenstates of the system. However, as we show here, the set of disorder realization where such a resonance-induced breakdown of Eq. (B17) occurs has measure zero in the thermodynamic limit. In this way, Eqs. (B17) and (B18) hold for *almost all* disorder realizations, in the thermodynamic limit.

To establish Eqs. (B17) and (B18), we first consider the case $\ell = 1$ (i.e., we establish the relationships for each single-particle Floquet eigenstate). Subsequently, in a stepwise fashion, we generalize this result to states with $\ell$ particles, for each $\ell = 2, \ldots k$.

### a. Single-particle eigenstates

Here we establish the relationships in Eqs. (B17) and (B18) for the single-particle case. We assume that $k$-particle localization is robust to perturbations, and thus $\tilde{U}$ also describes a $k$-particle localized system (we assume $k \geq 1$). Thus, in particular, each single-particle eigenstate $|\tilde{\Psi}\rangle$ of $\tilde{U}$ has its full support within a finite disk-shaped region $S$ of linear dimension $d$, with the probability of finding the particle a distance $s$ outside $S$ decaying as $e^{-s/\xi_l}$.

Due to its finite region of support, each single-particle eigenstate of $\tilde{U}$, $|\tilde{\Psi}\rangle$, may only overlap significantly with Floquet eigenstates whose corresponding LIOM centers are located within a distance $\sim \xi_l$ from $S$. To exploit this fact, we introduce a system-size dependent length scale $d \gg \xi_l$, which acts as an effective length cutoff for the region of support of a LIOM. The length $d$ must be much smaller than $L$, but can otherwise be taken to be arbitrarily large, as long as $d/L$ vanishes in the thermodynamic

limit. From the considerations above it follows that $|\tilde{\Psi}\rangle$ only overlaps with the finite number Floquet eigenstates, $|\Psi_{\alpha_1}\rangle \ldots |\Psi_{\alpha_{N_1}}\rangle$, whose LIOM centers are located within a distance $d$ from $S$, (up to a correction exponentially small in $d/\xi_l$:

$$\sum_{n=1}^{N_1} |\langle \Psi_{\alpha_n}|\tilde{\Psi}\rangle|^2 = 1 + \mathcal{O}(e^{-d/\xi_l}). \qquad \text{(B19)}$$

For the purposes of the following, it is convenient to order the indices $n$ according to the value of the overlap, such that $|\langle \Psi_{\alpha_1}|\tilde{\Psi}\rangle|^2 \geq |\langle \Psi_{\alpha_2}|\tilde{\Psi}\rangle|^2 \geq \ldots \geq |\langle \Psi_{\alpha_{N_1}}|\tilde{\Psi}\rangle|^2$. Note that the sequence of LIOM indices $\alpha_1 \ldots \alpha_{N_1}$ depends on the choice of $|\tilde{\Psi}\rangle$; this dependence is taken to be implicit below, for the sake of brevity.

We now show that $|\tilde{\Psi}\rangle$ only overlaps significantly with *one* of the eigenstates $|\Psi_{\alpha_1}\rangle \ldots |\Psi_{\alpha_{N_1}}\rangle$, while the total weight from all other eigenstates gives a negligible contribution. To show this, note that $|\Psi_{\alpha_n}\rangle$ and $|\tilde{\Psi}\rangle$ are eigenstates of $U$ and $\tilde{U}$, respectively, and hence

$$\langle \Psi_{\alpha_n}|\tilde{\Psi}\rangle = \frac{\langle \Psi_{\alpha_n}|U^\dagger \tilde{U} - 1|\tilde{\Psi}\rangle}{e^{-i(\tilde{E} - E_{\alpha_n})T} - 1}, \qquad \text{(B20)}$$

where $\tilde{E}$ is the quasienergy associated with $|\tilde{\Psi}\rangle$. Since $|\tilde{\Psi}\rangle$ is exponentially well localized within $S$, Eq. (B16) implies that $|\langle \Psi_{\alpha_n}|U^\dagger \tilde{U} - 1|\tilde{\Psi}\rangle| \lesssim JTA_S/L^2$. Moreover, $|e^{-i(\tilde{E}-E_{\alpha_n})T} - 1| \leq |\tilde{E} - E_n|T$, where the norm $|\cdot|$ is defined modulo $2\pi/T$, i.e. $|E| \equiv \min_z |E + 2\pi z/T|$. Combining these two inequalities with Eq. (B20), we find

$$|\langle \Psi_{\alpha_n}|\tilde{\Psi}\rangle| \lesssim \frac{JA_S/L^2 T}{|\tilde{E} - E_{\alpha_n}|}. \qquad \text{(B21)}$$

We now consider two implications of the above inequality. Firstly, Eq. (B19) implies $|\langle \Psi_{\alpha_1}|\tilde{\Psi}\rangle|^2 \gtrsim 1/N_1 - \mathcal{O}(e^{-d/\xi_l})$ (c.f. the labelling of the states $\{|\Psi_{\alpha_n}\rangle\}$). Thus,

$$|\tilde{E} - E_{\alpha_1}| \lesssim \sqrt{N_1} JA_S/L^2 T. \qquad \text{(B22)}$$

Secondly, we note that, for a random choice of $|\tilde{\Psi}\rangle$, the typical spacing between the $N_1$ quasienergy levels $\{E_n\}$ is of order $\Delta E \sim \mathcal{W}/N_1$, where $\mathcal{W}$ denotes the width of the single-particle quasienergy spectrum (when the quasienergy spectrum has no gaps, $\mathcal{W} = 2\pi/T$). In this case, only one of the quasienergies $\{E_{\alpha_n}\}$ (namely $E_{\alpha_1}$) is close enough to $\tilde{E}$ for Eq. (B21) to allow a significant value of $\langle \Psi_n|\tilde{\Psi}\rangle$. Thus, $|\tilde{\Psi}\rangle \approx |\Psi_1\rangle$ for a typical choice of $|\tilde{\Psi}\rangle$.

We now prove that $|\tilde{\Psi}\rangle \approx |\Psi_1\rangle$ for *any* choice of $|\tilde{\Psi}\rangle$ in the system (except for a measure-zero set of disorder realizations in the thermodynamic limit). To establish this result, we first note

$$|E_n - \tilde{E}| \geq |E_{\alpha_n} - E_{\alpha_1}| - |\tilde{E} - E_{\alpha_1}|. \qquad \text{(B23)}$$

We now establish a lower bound for $|E_{\alpha_n} - E_{\alpha_1}|$, using the fact the quasienergy levels of nearby states $E_{\alpha_1}$

and $E_{\alpha_n}$ repel each other, and that $|\tilde{E} - E_{\alpha_1}|$ satisfies the bound of Eq. (B22). Specifically, note that the Floquet eigenstates $|\Psi_1\rangle$ and $|\Psi_n\rangle$ have their support within a distance $\lesssim d$ from each other. The quasienergies $E_{\alpha_1}$ and $E_{\alpha_n}$ are hence subject to local level repulsion when the quasienergy difference $\delta E \equiv |E_n - E_1|$ is much smaller than the scale of matrix elements between them with respect to the kinetic part of the Hamiltonian (i.e. $\delta E \ll Je^{-d/\xi_l}$). In the limit where $\delta E \ll Je^{-d/\xi_l}$, the probability distribution $p(\delta E)$ for $\delta E$ should thus resemble the Wigner-Dyson distribution for the Circular unitary ensemble (CUE) [67]:

$$p(\delta E) = \frac{T^3}{\pi}\delta E^2 + \mathcal{O}(\delta E^4). \qquad (B24)$$

Using the above result, we now compute the expected number of pairs of nearby single-particle eigenstates $|\Psi_{\alpha_i}\rangle$ and $|\Psi_{\alpha_j}\rangle$ in the entire system, for which $|E_{\alpha_i} - E_{\alpha_j}|$ is smaller than some given (small) value $\delta E_0$. Here "nearby" refers to the eigenstates $|\Psi_{\alpha_i}\rangle$ and $|\Psi_{\alpha_j}\rangle$ having their centers located within a distance $\sim d$ from each other, such that they may potentially overlap with the same eigenstate of $\tilde{U}$. Noting that there are $\mathcal{O}(L^2 N_1/2a^2)$ distinct pairs of nearby eigenstates (where $a$ denotes the lattice constant), we have

$$N(\delta E_0) = \frac{L^2 d^2}{2a^4}\int_0^{\delta E_0} d\delta E\, p(\delta E). \qquad (B25)$$

where we used $N_1 \sim (d/a)^2$. Thus, in the limit where $\delta E_0 \ll Je^{-d/\xi_l}$,

$$N(\delta E_0) = \frac{L^2 d^2(\delta E_0 T)^3}{6\pi a^4}. \qquad (B26)$$

We recall we may take $d$ arbitrarily large as long as $d/L \to 0$ in the thermodynamic limit. In the following, it is convenient to let $d$ scale with system-size as $d \sim \frac{1}{2}\xi_\ell \log(L/a)$ such that $\mathcal{O}(e^{-d/\xi_l}) \sim \mathcal{O}(L^{-1/2})$. (Note that this choice is not unique; other scaling behaviors of $d$ can be used in the discussion below). Since with this choice of $d$, $Je^{-d/\xi} \gg a/LT$ in the thermodynamic limit [such that Eq. (B26) applies to $\delta E_0 = a/LT$], we conclude that

$$\lim_{L \to \infty} N(a/LT) = 0. \qquad (B27)$$

We conclude that, in the thermodynamic limit, there are zero pairs of Floquet eigenstates $|\Psi_{\alpha_i}\rangle$ and $|\Psi_{\alpha_j}\rangle$ with LIOM centers within a distance $d \sim \frac{1}{2}\xi_\ell \log(L/a)$ from each other whose quasienergies differ by less than $\frac{a}{LT}$ (except for in a measure zero set of disorder realizations). We conclude, in the thermodynamic limit, and for *any* choice of $|\tilde{\Psi}\rangle$,

$$|E_{\alpha_1} - E_{\alpha_n}| > \frac{a}{LT} \qquad (B28)$$

for all but a measure zero set of disorder realizations.

Using Eq. (B23) along with the fact that $|\tilde{E} - E_{\alpha_1}|$ is subleading in $L$ compared to the above bound for $|E_{\alpha_n} - E_{\alpha_1}|$, we find, for $n \geq 2$, $|\tilde{E} - E_{\alpha_n}| > \frac{a}{LT}$. Thus, for all but a measure zero set of disorder realizations, it holds that, for *each* choice of $|\tilde{\Psi}\rangle$,

$$|\langle\Psi_{\alpha_n}|\tilde{\Psi}\rangle| < \frac{A_S JT}{aL} \quad \text{for} \quad n \geq 2. \qquad (B29)$$

Using this result in Eq. (B19), we find

$$1 - |\langle\Psi_{\alpha_1}|\tilde{\Psi}\rangle|^2 < \frac{N_1 A_S^2 J^2 T^2}{a^2 L^2} + \mathcal{O}(e^{-d/\xi_l}). \qquad (B30)$$

Recall that we take $d \sim \frac{1}{2}\xi_l \log(L/a)$. Hence the first term above is subleading in the thermodynamic limit, and we obtain

$$|\langle\Psi_{\alpha_1}|\tilde{\Psi}\rangle|^2 = 1 + \mathcal{O}(L^{-1/2}). \qquad (B31)$$

(See Footnote 64). This concludes the proof of Eq. (B17) for the single-particle case, when we assign the label $\alpha_1$ to $|\tilde{\Psi}\rangle$.

To establish Eq. (B18) for the single-particle case, we note from Eq. (B21) [with the labelling for $|\tilde{\Psi}\rangle$ introduced above] that, for each choice of $\alpha$,

$$|\tilde{E}_\alpha - E_\alpha| \lesssim \frac{JA_S/L^2}{|\langle\tilde{\Psi}_\alpha|\Psi_\alpha\rangle|}. \qquad (B32)$$

Since $|\langle\tilde{\Psi}_\alpha|\Psi_\alpha\rangle| \approx 1$, and $A_S \sim d^2$, we conclude $|\tilde{E}_\alpha - E_\alpha| \lesssim Jd^2/L^2 \sim \mathcal{O}(L^{-2})$ (see Footnote [64]). This is what we wanted to show.

### b. Two-particle eigenstates

Having established Eq. (B17) for single-particle Floquet eigenstates, we now show that it also holds for all two-particle eigenstates (provided the system is $k$-particle localized for some $k \geq 2$). In order to do this, we consider a two-particle Floquet eigenstate $|\tilde{\psi}\rangle$ of the one-flux system, with quasienergy $\tilde{E}$. Since the one-flux system is $k$-particle localized (for $k \geq 2$), the two-particle eigenstates of $\tilde{U}$ possess a LIOM structure. In the Floquet eigenstate $|\tilde{\psi}\rangle$, two of the LIOMs of $\tilde{U}$, $\tilde{n}_1$ and $\tilde{n}_2$, are thus "excited" (i.e. $\tilde{n}_\alpha|\tilde{\psi}\rangle = |\tilde{\psi}\rangle$ for $\alpha = 1,2$, while $\tilde{n}_\alpha|\tilde{\psi}\rangle = 0$ for $\alpha \neq 1,2$). In the following, we divide our argumentation into two cases, depending on whether or not the LIOMs $\tilde{n}_1$ and $\tilde{n}_2$ are located within a distance $d$ from each other, where $d$ denotes the arbitrary length scale cutoff for each LIOM's region of support introduced in Sec. B 4 a.

*Nearby LIOMs* — When the centers of the two "excited" LIOMs $\tilde{n}_1$ and $\tilde{n}_2$ in the state $|\tilde{\psi}\rangle$ are separated by a distance less than $d$, a two-particle Floquet eigenstate $|\Psi_{\alpha\beta}\rangle$ of the zero-flux system may only significantly overlap with $|\tilde{\Psi}\rangle$ if the corresponding excited (nonperturbed)

LIOMs $\hat{n}_\alpha$ and $\hat{n}_\beta$ are located within a distance $d$ from the centers of $\tilde{n}_1$ and $\tilde{n}_2$. As a result, there are only of order $N_2 \sim \binom{2d^2/a^2}{2}$ choices of distinct LIOMs $\alpha, \beta$ for which $|\Psi_{\alpha\beta}\rangle$ can significantly overlap with $|\tilde{\Psi}\rangle$.

Using the same arguments as for the single particle case (Sec. B 4 a) one can show that, for all but a measure-zero set of disorder realizations in the thermodynamic limit, there exists a unique two-particle eigenstate $|\Psi_{\alpha\beta}\rangle$ of $U$ for each two-particle eigenstate $|\tilde{\Psi}\rangle$ of $\tilde{U}$ such that (up to a gauge transformation)

$$|\tilde{\Psi}\rangle = |\Psi_{\alpha\beta}\rangle + \mathcal{O}\left(L^{-1/2}\right), \qquad (B33)$$

and

$$\tilde{E} = E_{\alpha\beta} + \mathcal{O}\left(L^{-2}\right). \qquad (B34)$$

*Separated LIOMs* — Next, we consider the case where the two excited LIOMs $\tilde{n}_1$ and $\tilde{n}_2$ are separated by a distance $\Delta r$ larger than $d$. In this case, the LIOM structure of the Floquet operator $\tilde{U}$ [Eq. (1) in the main text] implies that, up to an exponentially small correction in the distance $\Delta r / \xi_l$, $|\tilde{\Psi}\rangle$ may be written as a direct product of two single-particle eigenstates $|\tilde{\Psi}_\alpha\rangle$ and $|\tilde{\Psi}_\beta\rangle$. Here $\alpha$ and $\beta$ refer to the labeling of the single-particle eigenstates of $\tilde{U}$ that was established in the previous subsection. Letting $S_\alpha$ and $S_\beta$ denote the two non-overlapping regions of linear dimension $d$ where the states $|\tilde{\Psi}_\alpha\rangle$ and $|\tilde{\Psi}_\beta\rangle$ respectively have their support (up to a correction exponentially small in $d/\xi_l$), we have [68]:

$$|\tilde{\Psi}\rangle = |\tilde{\Psi}_\alpha\rangle_{S_\alpha} \otimes |\tilde{\Psi}_\beta\rangle_{S_\beta} \otimes |0\rangle + \mathcal{O}(e^{-d/\xi_l}). \qquad (B35)$$

where we used $\Delta r > d$. Here $|\Psi\rangle_S$ denotes the restriction of the state $|\Psi\rangle$ to the Fock space of the region $S$ (defined from the projection of $|\Psi\rangle$ into the subspace with no particles outside region $S$). The state $|0\rangle$ refers to the vacuum in the complementary region to $S_\alpha$ and $S_\beta$. Since the two particles in the state $|\tilde{\Psi}\rangle$ are separated by a distance much larger than $d$, the regions $S_\alpha$ and $S_\beta$ do not overlap.

We recall that Eq. (B17) was already proven to hold for the single-particle case. Thus $|\tilde{\Psi}_\alpha\rangle$ (the eigenstate in the presence of one flux quantum piercing the system) is approximately identical to a single-particle eigenstate $|\Psi_\alpha\rangle$ of the zero-flux system's Floquet operator $U$ (for all but a measure zero set of disorder realizations). Specifically, up to a gauge transformation, $|\tilde{\Psi}_\alpha\rangle = |\Psi_\alpha\rangle + \mathcal{O}(L^{-2})$. The eigenstate $|\Psi_\alpha\rangle$ moreover has its full support in the same region $S_\alpha$ as $|\tilde{\Psi}_\alpha\rangle$, up to a correction exponentially small in $d/\xi_l$. Letting $V_\alpha$ be the unitary operator that generates the transformation to the gauge in which Eq. (B17) holds for $|\tilde{\Psi}_\alpha\rangle$, we have

$$|\tilde{\Psi}_\alpha\rangle_{S_\alpha} = V_\alpha |\Psi_\alpha\rangle_{S_\alpha} + \mathcal{O}(L^{-1/2}), \qquad (B36)$$

where we used that we may take $d \sim \frac{1}{2}\xi_l \log(L/a)$, such that the correction $\mathcal{O}(e^{-d/\xi})$ scales with system size

as $L^{-1/2}$ in the thermodynamic limit. Using the relation (B36) for the states $|\tilde{\Psi}_\alpha\rangle_{S_\alpha}$ and $|\tilde{\Psi}_\beta\rangle_{S_\beta}$ in Eq. (B35), we hence obtain

$$|\tilde{\Psi}\rangle = V_\alpha V_\beta |\Psi_\alpha\rangle_{S_\alpha} \otimes |\Psi_\beta\rangle_{S_\beta} \otimes |0\rangle + \mathcal{O}(L^{-1/2}). \qquad (B37)$$

Due to the LIOM structure of the Floquet operator $U$ (Eq. (1) in the main text), $|\Psi_\alpha\rangle_{S_\alpha} \otimes |\Psi_\beta\rangle_{S_\beta} \otimes |0\rangle$ is identical to the Floquet eigenstate $|\Psi_{\alpha\beta}\rangle$ of the zero-flux system, up to a correction of order $e^{-d/\xi_l}$. Since the product of the two gauge transformations $V_\alpha$ and $V_\beta$ is itself a gauge transformation, we thus conclude that, up to a gauge transformation:

$$|\tilde{\Psi}\rangle = |\Psi_{\alpha\beta}\rangle + \mathcal{O}(L^{-1/2}). \qquad (B38)$$

The two cases we considered above show that, in the thermodynamic limit, and for all but a measure zero set of disorder realizations, *each* two-particle eigenstate $|\tilde{\Psi}\rangle$ of $\tilde{U}$ is identical to a unique eigenstate of $U$, up to a gauge transformation, and a correction of order $\mathcal{O}(L^{-1/2})$. We may thus label the two-particle eigenstates of $\tilde{U}$ such that Eqs. (B17) and (B18) hold with $\ell = 2$, and for each choice of the LIOM indices $\alpha_1$ and $\alpha_2$.

### c. $\ell$-particle-eigenstates

We finally consider the general case of an $\ell$-particle eigenstate $|\tilde{\Psi}\rangle$ of $\tilde{U}$, where $\ell$ is smaller than or equal to the system's degree of localization, $k$. For this situation, we can apply the same structure of arguments as for the two-particle case: due to the LIOM structure of the one-flux Floquet operator $\tilde{U}$, each $\ell$-particle state is constructed by "exciting" $\ell$ LIOMs $\tilde{n}_1 \dots \tilde{n}_\ell$. We split our line of arguments into two cases, depending on whether or not the LIOMs $\tilde{n}_1 \dots \tilde{n}_\ell$ can be divided into clusters separated from each other by distances greater than $d$.

In the case where the excited LIOMs *can* be divided into clusters in the way above, $|\tilde{\psi}\rangle$ can be written as a direct product of eigenstates of $\tilde{U}$ with fewer than $k$ particles, up to a correction of order $e^{-d/\xi_l}$. Following the same line of arguments as for the analogous two-particle case, the relationships (B17) and (B18) can then be demonstrated to hold for this class of eigenstates using the fact that Eq. (B17) and (B18) hold for eigenstates with fewer than $\ell$ particles.

In the case where all LIOMs are located in the same cluster, we note that $|\tilde{\psi}\rangle$ only significantly overlaps with eigenstates $\{|\Psi_{\alpha_1 \dots \alpha_\ell}\rangle\}$ where the centers of all the LIOMs $\hat{n}_{\alpha_1} \dots \hat{n}_{\alpha_\ell}$ are located in the region $S$, consisting of all sites with a distance $d$ from any of the excited LIOM's $\tilde{n}_1 \dots \tilde{n}_\ell$. There only exist a finite number of eigenstates $N_\ell$ with this property. Specifically, $N_\ell \lesssim \binom{\ell d^2/a^2}{\ell}$ counts the number of distinct configurations of $k$ LIOMs $\hat{n}_{\alpha_1} \dots \hat{n}_{\alpha_\ell}$ whose centers are located within $S$. Crucially, $N_\ell$ only depends on the number of particles, $\ell$, and $d$, and is independent of system size.

Using the same arguments as for the single-particle case, we then find that, for all but a measure zero set of disorder realizations in the thermodynamic limit, there exists a unique eigenstate $|\Psi_{\alpha_1\ldots\alpha_\ell}\rangle$ of $U$ such that (up to a gauge transformation),

$$|\tilde{\Psi}\rangle = |\Psi_{\alpha_1\ldots\alpha_\ell}\rangle + \mathcal{O}(L^{-1/2}), \tag{B39}$$

where, as we described in the beginning of this Appendix, $\mathcal{O}(L^{-p})$ denotes term scaling with system size as $L^{-p}$ in the thermodynamic limit (see Footnote [64]). In addition, when the LIOMs are located within a distance $d$ from the same point,

$$\tilde{E} = E_{\alpha_1\ldots\alpha_\ell} + \mathcal{O}\left(L^{-2}\right). \tag{B40}$$

Thus, Eqs. (B17) and (B18) hold for the $\ell$-particle case in the thermodynamic limit, for any $\ell = 1,\ldots k$.

### 5. Relationship between magnetization density and quasienergy

Having established the auxiliary results in Secs. B 1-B 4, we are now ready to prove Eq. (B1), which is the first main goal of this appendix. To recapitulate, we seek to show that, for each $\ell$-particle Floquet eigenstate, $|\psi_n\rangle$ with quasienergy $\varepsilon_n$, the associated quasienergy for the one-flux system, $\tilde{\varepsilon}_n$ (see Sec. B 4 for details), satisfies

$$\tilde{\varepsilon}_n = \varepsilon_n + B_0\langle\psi_n|\bar{M}|\psi_n\rangle + \mathcal{O}(L^{-5/2}), \tag{B41}$$

where $\bar{M}$ denotes the time-averaged magnetization operator (see Sec. III B 1 of the main text), and, as in Sec. B 4 above, $\mathcal{O}(L^{-p})$ denotes a correction of order $\lambda L^{-p}$ or less, where $\lambda$ is some system-size independent energy scale that does not play a role for our discussion.

In this step of the derivation it is useful to define a region of support, $S_n$, for each Floquet eigenstate $|\psi_n\rangle$. Specifically, for each Floquet eigenstate, $|\psi_n\rangle$, and for some length scale $d \ll L$, we let $S_n$ denote the smallest region of the lattice that ensures the centers of all nonzero LIOMs in the state $|\psi_n\rangle$, $\alpha_1\ldots\alpha_\ell$, are located within a distance $d$ from the boundary of $S_n$. The region of support $S_n$ may consist of one or several disconnected disk-shaped subregions of linear dimension $d$, and has area $A_{S_n} \leq \pi\ell d^2$. As in Sec. B 4, when taking the thermodynamic limit $L \to \infty$ in the following, we let $d$ increase logarithmically with system size as $d \sim \frac{1}{2}\xi_l\log(L/a)$.

To establish Eq. (B41), for a given Floquet eigenstate $|\psi_n\rangle$, we let $\tilde{U}$ be the one-flux Floquet operator in a gauge where Eq. (B16) holds within $S_n$, and let $|\tilde{\psi}_n\rangle$ denote the eigenstate of $\tilde{U}$ corresponding to $|\psi_n\rangle$ through Eq. (B17). Noting that $|\psi_n\rangle$ and $|\tilde{\psi}_n\rangle$ are eigenstates of $U$ and $\tilde{U}$, respectively, we have

$$\langle\psi_n|U^\dagger\tilde{U}|\tilde{\psi}_n\rangle = e^{-i(\tilde{\varepsilon}_n-\varepsilon_n)T}\langle\psi_n|\tilde{\psi}_n\rangle. \tag{B42}$$

At the same time, the left-hand side above can be written [see Eq. (B13)],

$$\langle\psi_n|U^\dagger\tilde{U}|\tilde{\psi}_n\rangle = \langle\psi_n|\tilde{\psi}_n\rangle$$
$$- i\int_0^T dt\langle\psi_n|U^\dagger(t)\delta H(t)\tilde{U}(t)|\tilde{\psi}_n\rangle. \tag{B43}$$

We now seek to rewrite the second term above to a form which only relies quantities of the (unperturbed) zero-flux system. Using that $|\tilde{\psi}_n\rangle = |\psi_n\rangle + \mathcal{O}(L^{-1/2})$ [Eq. (B17)], and that $U|\psi\rangle = \tilde{U}|\psi\rangle + \mathcal{O}(L^{-2})$ for normalized states $|\psi\rangle$ that are exponentially localized within $S_n$ (such as $|\psi_n\rangle$), we find

$$\tilde{U}(t)|\tilde{\psi}_n\rangle = U(t)|\psi_n\rangle + \mathcal{O}(L^{-1/2}). \tag{B44}$$

We recall from Eq. (B15) that $\|\delta H(t)U(t)|\psi_n\rangle\| \sim \mathcal{O}(L^{-2})$, such that, for any state $|\psi\rangle$, $|\langle\psi_n|U^\dagger(t)\delta H(t)|\psi\rangle| \lesssim \mathcal{O}(L^{-2})\||\psi\rangle\|$. Combining this with Eqs. (B43) and (B44), we find

$$e^{-i(\tilde{\varepsilon}_n-\varepsilon_n)T}\langle\psi_n|\tilde{\psi}_n\rangle = \langle\psi_n|\tilde{\psi}_n\rangle$$
$$- i\int_0^T dt\langle\psi_n|U^\dagger(t)\delta H(t)U(t)|\psi_n\rangle + \mathcal{O}(L^{-5/2}). \tag{B45}$$

We finally note that $\langle\psi_n|\tilde{\psi}_n\rangle = 1 + \mathcal{O}(L^{-2})$. Dividing through with a factor of $\langle\psi_n|\tilde{\psi}_n\rangle$, and again using that $\|\langle\psi_n|U^\dagger(t)\delta H(t)\| \sim \mathcal{O}(L^{-2})$, we hence obtain

$$e^{-i(\tilde{\varepsilon}_n-\varepsilon_n)T} = 1 - i\int_0^T dt\langle\psi_n|U^\dagger(t)\delta H(t)U(t)|\psi_n\rangle$$
$$+ \mathcal{O}(L^{-5/2}). \tag{B46}$$

Expanding the left-hand side to first order in $\tilde{\varepsilon}_n - \varepsilon_n$, and using $\tilde{\varepsilon}_n - \varepsilon_n \sim \mathcal{O}(L^{-2})$ [see Eq. (B40)], we obtain

$$\tilde{\varepsilon}_n - \varepsilon_n = \frac{1}{T}\int_0^T dt\langle\psi_n|U^\dagger(t)\delta H(t)U(t)|\psi_n\rangle + \mathcal{O}(L^{-5/2}). \tag{B47}$$

Having expressed $\tilde{\varepsilon}_n - \varepsilon_n$ purely in terms of quantities of the zero-flux system, we now relate the first term on the right-hand side above to the time-averaged magnetization in the state $|\psi_n\rangle$. To this end, we use the explicit form of $H(t)$ we assumed in Eq. (B3) (similar arguments apply to more general Hamiltonians), finding

$$\delta H(t) = i\sum_{ij}\theta_{ij}J_{ij}(t)\hat{c}_i^\dagger\hat{c}_j + \delta H^{(2)}(t), \tag{B48}$$

where $\delta H^{(2)}(t) = \sum_{ij}[e^{i\theta_{ij}} - 1 - i\theta_{ij}]J_{ij}(t)\hat{c}_i^\dagger\hat{c}_j$ and $\{\theta_{ij}\}$ denote the Peierls phases induced by the uniform magnetic field $B_0$. We identify $-i[J_{ij}(t)\hat{c}_i^\dagger\hat{c}_j - J_{ji}(t)\hat{c}_j^\dagger\hat{c}_i]$ as the bond current operator on the bond from site $j$ to site $i$, $\hat{I}_b(t)$, and $\theta_{ij}$ as the associated Peierls phase (see also Footnote 49). Hence, $i\sum_{ij}\theta_{ij}J_{ij}(t)\hat{c}_i^\dagger\hat{c}_j = -\sum_b\theta_bI_b(t)$.

To establish a bound for the term in Eq. (B47) originating from $\delta H^{(2)}(t)$, we note that $\theta_{ij} \sim \mathcal{O}(L^{-2})$ within

the region of support of the state $|\psi_n\rangle$, $S_n$. Hence $[e^{-i\theta_{ij}} - (1 - i\theta_{ij})] \sim \mathcal{O}(L^{-4})$ for sites $i, j$ within $S_n$. As a result, $\|\delta H^{(2)}(t)|\psi_n\rangle\| \sim \mathcal{O}(L^{-4})$. Thus, we obtain

$$\tilde{\varepsilon}_n - \varepsilon_n = -\sum_b \theta_b \int_0^T \frac{\mathrm{d}t}{T} \langle\psi_n|U^\dagger(t)\hat{I}_b(t)U(t)|\psi_n\rangle + \mathcal{O}(L^{-5/2}). \tag{B49}$$

Using that in a Floquet eigenstate the time-averaged expectation value over one period equals the long-time average, we obtain

$$\tilde{\varepsilon}_n - \varepsilon_n = -\sum_b \theta_b \langle\psi_n|\bar{I}_b|\psi_n\rangle + \mathcal{O}(L^{-5/2}), \tag{B50}$$

where $\bar{\mathcal{O}}$ denotes the long-time average in the Heisenberg picture [see Eq. (3)]. Retracing the arguments in the main text that lead from Eq. (12) to Eq. (13), we find $\sum_b \theta_b \bar{I}_b = \bar{M}B_0$. Thus, we conclude

$$\tilde{\varepsilon}_n - \varepsilon_n = -\langle\psi_n|\bar{M}|\psi_n\rangle B_0 + \mathcal{O}(L^{-5/2}). \tag{B51}$$

This establishes Eq. (B41), which was the first goal of this Appendix.

### 6. Vanishing sum of corrections

As the final step of this Appendix, we now show that the corrections to Eq. (B51) (which individually scale with system size, $L$, as $L^{-4}$), approximately cancel out when summed over all quasienergy levels in the $\ell$-particle subspace, yielding an *exponentially* suppressed correction:

$$\sum_n (\tilde{\varepsilon}_n - \varepsilon_n) = -\sum_n B_0 \langle\psi_n|\bar{M}|\psi_n\rangle + \mathcal{O}(e^{-L/\xi_l}), \tag{B52}$$

To establish Eq. (B52), it is convenient to first express Eq. (B51) in terms of the magnetization densities on each plaquette, $\{\bar{m}_p\}$ by using $\bar{M} = \sum_p a^2 \bar{m}_p$:

$$\tilde{\varepsilon}_n - \varepsilon_n = -\sum_p a^2 B_0 \langle\psi_n|\bar{m}_p|\psi_n\rangle + \mathcal{O}(L^{-5/2}). \tag{B53}$$

To obtain Eq. (B52) from the above result, we exploit the LIOM decomposition of the quasienergy levels in terms of the quasienergy coefficients $\varepsilon_{\alpha_1}, \varepsilon_{\alpha_1\alpha_2}, \ldots$ [Eq. (1)], and the analogous decomposition time-averaged magnetization density in term of the magnetization coefficients $\bar{m}_{\alpha_1}^p, \bar{m}_{\alpha_1\alpha_2}^p, \ldots$ [Eq. (19)]. By inserting these expansions into Eq. (B53) and using that Eq. (B53) holds for each Floquet eigenstate with up to $\ell$ particles (i.e., for each combination of up to $\ell$ excited LIOMs), one can verify that, for each choice of $\ell$ LIOMs, $\alpha_1 \ldots \alpha_\ell$,

$$\tilde{\varepsilon}_{\alpha_1\ldots\alpha_\ell} - \varepsilon_{\alpha_1\ldots\alpha_\ell} = -B_0 \sum_p a^2 \bar{m}_{\alpha_1\ldots\alpha_\ell}^p + \mathcal{O}(L^{-5/2}). \tag{B54}$$

We now seek to compute the sum the left hand side above over all $\binom{D_1}{\ell}$ distinct combinations of $\ell$ LIOMs, where $\binom{a}{b}$ denotes the binomial coefficient and $D_1 = L^2/a^2$ the dimension of the system's single-particle subspace. Specifically, we seek to compute

$$\kappa_\ell \equiv \sum_{\alpha_1\ldots\alpha_\ell} (\tilde{\varepsilon}_{\alpha_1\ldots\alpha_\ell} - \varepsilon_{\alpha_1\ldots\alpha_\ell}). \tag{B55}$$

Since $\bar{m}_{\alpha_1\ldots\alpha_\ell}^p$ and $\varepsilon_{\alpha_1\ldots\alpha_\ell}$ may only be nonzero when the LIOMs $\alpha_1\ldots\alpha_\ell$ are located within a distance $\sim \xi_l$ from each other, there are of order $L^2/a^2$ combinations of $\ell$ LIOMs for which $\bar{m}_{\alpha_1\ldots\alpha_\ell}^p$ and $\varepsilon_{\alpha_1\ldots\alpha_\ell}$ may be significant. Summing Eq. (B54) over these $\mathcal{O}(L^2)$ combinations, we obtain

$$\kappa_\ell = -B_0 \sum_p \sum_{\alpha_1\ldots\alpha_\ell} a^2 \bar{m}_{\alpha_1\ldots\alpha_\ell}^p + \mathcal{O}(L^{-1/2}). \tag{B56}$$

To obtain $\kappa_\ell$, we use $\kappa_1, \ldots \kappa_\ell$ to express the sum of $\tilde{\varepsilon}_n - \varepsilon_n$ over all $\ell$-particle quasienergy levels. An argument similar to the one made in Sec. III C shows that the sum of $\tilde{\varepsilon}_n - \varepsilon_n$ over all $\ell$-particle quasienergy levels yields

$$\sum_n (\tilde{\varepsilon}_n - \varepsilon_n) = \sum_{\ell'=1}^\ell \binom{D_1 - \ell'}{\ell - \ell'} \sum_{\alpha_1\ldots\alpha_{\ell'}} \kappa_\ell \tag{B57}$$

Note, in particular, that the sum of $\tilde{\varepsilon}_n - \varepsilon_n$ over all single-particle quasienergy levels is identical to $\kappa_1$. Using that $\sum_n(\tilde{\varepsilon}_n - \varepsilon_n)$ must be quantized an integer multiple of $2\pi/T$ (see Sec. III B 1 in the main text) along with an inductive argument similar to the one below Eq. (18) in the main text, we conclude that $\kappa_\ell$ must be an integer multiple of $2\pi/T$ for each $\ell \leq k$.

Using inductive arguments similar to the ones employed above, using that $\mathrm{Tr}_{\ell'}\bar{m}_p = \mathrm{Tr}_{\ell'}\bar{m}_q$ for any two plaquettes $p, q$ in the lattice, for any $\ell' = 1, \ldots \ell$, it follows that,

$$\sum_{\alpha_1\ldots\alpha_k} \bar{m}_{\alpha_1\ldots\alpha_k}^p = \sum_{\alpha_1\ldots\alpha_\ell} \bar{m}_{\alpha_1\ldots\alpha_\ell}^q. \tag{B58}$$

Using this result in Eq. (B56) along with $L^2 B_0 = 2\pi$, we thus find, for any given plaquette $p_0$ in the lattice,

$$\kappa_\ell = 2\pi \sum_{\alpha_1\ldots\alpha_\ell} \bar{m}_{\alpha_1\ldots\alpha_\ell}^{p_0} + \mathcal{O}(L^{-1/2}). \tag{B59}$$

We now consider how the right- and left-hand sides differ from their values in the thermodynamic limit, $L \to \infty$. Firstly, $\bar{m}_{\alpha_1\ldots\alpha_\ell}^{p_0}$ is exponentially suppressed for all but the $\binom{\xi_l^2/a^2}{\ell}$ choices of $\ell$ LIOMs where all LIOM centers are all located within a distance $\sim \xi_l$ from plaquette $p_0$. Hence $\sum_{\alpha_1\ldots\alpha_\ell} \bar{m}_{\alpha_1\ldots\alpha_\ell}^{p_0}$ only depends on the details of the system near plaquette $p$, and therefore can only differ from its value in the thermodynamic limit by an amount of order $e^{-L/\xi_l}$. From below Eq. (B57) we recall that $\kappa_\ell$ is exactly quantized as an integer multiple of $2\pi/T$. Moreover, Eq. (B59) shows that $\kappa_\ell$ can

only differ from $\sum_{\alpha_1...\alpha_\ell} \bar{m}^{p_0}_{\alpha_1...\alpha_\ell}$ by an amount of order $\mathcal{O}(L^{-2})$. Hence, when $L \gg \xi_l$, $\kappa_\ell$ must be exactly identical to its value in thermodynamic limit. We conclude that $\delta_\ell \equiv \kappa_\ell - 2\pi \sum_{\alpha_1...\alpha_\ell} \bar{m}^{p_0}_{\alpha_1...\alpha_\ell}$ can only differ from its value in the thermodynamic limit by an amount of order $e^{-L/\xi_l}$. Since $\delta_\ell = 0$ in the thermodynamic limit, we find, for each plaquette in the system, $p_0$,

$$\sum_{\alpha_1...\alpha_\ell} (\varepsilon_{\alpha_1...\alpha_\ell} - 2\pi \bar{m}^{p_0}_{\alpha_1...\alpha_\ell}) = \mathcal{O}(e^{-L/\xi_l}). \qquad \text{(B60)}$$

Using $\bar{M} = \sum_p a^2 \bar{m}_p$ along with the LIOM decompositions in Eqs. (1) and (19), we conclude that Eq. (B52) holds. This was the goal of this subsection, and concludes this Appendix.