# Peer review of "Hierarchy of many-body invariants and quantized magnetization in anomalous Floquet insulators"

_SciPost Physics_

## Round 1 · Referee Report · Anonymous (Referee 1) · 2019-11-11

Strengths

See report

Weaknesses

See report

Report

The AFAI is a paradigmatic example of a non-trivial (non-interacting) phase of matter enabled by periodic driving. The phase is characterized by an edge current and a quantized bulk invariant related to the "magnetization" density. In this work, the authors attempt to generalize the bulk invariants of the AFAI to the interacting case, and argue that they get a hierarchy of bulk invariants and a richer structure than that of the non-interacting problem. This central result is certainly very interesting and deserves publication. However, several of the key steps in the arguments are currently unclear to me, and I am hoping the authors can clarify these prior to publication:

  1. The existence of an MBL stabilized interacting AFI is assumed as a starting point. However, as the authors acknowledge themselves, the stability of MBL in the presence of a thermalizing edge can be problematic. They attempt to sidestep this issue by working with periodic boundary conditions. However, it wasn't clear to me whether this actually solves the problem or just "hides" it. For example, in the ideal model of the AFAI, the Floquet operator over one period looks trivial with periodic boundary conditions --- but the non-trivial signatures of the phase can be gleaned by examining the micromotion within a period. Likewise, even if the Floquet unitary over one period is localized with PBCs for the AFI (ignoring avalanches etc.), is it clear that the micromotion within the period will not see signatures of delocalization and affect the invariants that the authors are trying to construct?

  2. One of the central steps is equation (5) which says that the current through a cut is only sensitive to plaquettes near the boundary of the cut. However, this does not address resonances in any way, which will inevitably be present in a large enough system and presumably invalidate equation (10). These will certainly be rare, but to what extent do they interfere with the `topological' characterization?

  3. I found the discussion below equation (10) very confusing. The authors discuss slowly perturbing some region R, and assert that this does not change the expectation values for $\overline{m_p}$ in the infinite size limit. They rely here on the assumption that $\overline{m_p}$ only depends on the region near $p$, and hence is affected by an exponentially small amount if it is located a distance $\sim O(L)$ from $R$. However, slowly changing $R$ should lead to all sorts of level crossings, resonances and global rearrangements in the system --- why should these leave $m_p$ unchanged?

The authors do mention such resonances later in Section IID when examining the response to a changing magnetic field. Here they seem to imply that the resonances can simply be "gauged away", which I found quite surprising -- how does a gauge choice get around the resonances that will certainly be created? Further, this mechanism (even if true), only seems to apply to a changing magnetic field while the discussion below (10) is seemingly supposed to hold for all weak deformations that preserve MBL. Surely not all of these lead to resonances that can be "gauged away"? Since this is one of the central steps of the proof, the authors should clarify these concerns.

  1. The authors make the point that the (new) higher order invariants only exist in models that cannot be continuously deformed to an AFAI by tuning down interactions. If this is the case, what is the basis for the assumption of MBL in such models? Are there any known examples of MBL systems --- especially those with a LIOM description, which the authors rely on --- that cannot be deformed to a non-interacting localized system? I think this is a very strong assumption that is largely left unjustified - and the central result in this paper (the existence of higher order invariants) turns on this.

It is also interesting that the numerical example presented is for the AFI that can be connected to the AFAI and not for the new "model" with higher-order invariants.

The authors do say at the end of Section III that the "new" model may not support MBL (and hence all the phenomenology discussed) and leave this an interesting open question. But with this left as an open question, I'm not sure what exactly the take home message of the paper is. The title talks about a `hierarchy of invariants', but the conclusion that these actually exist in any model needs the new class of models being discussed to (i) be MBL (ii) with a LIOM description and (iii) without a non-interacting AFAI limit. Without justifying (i)-(iii), the conclusion seems to me to not be warranted (this is even ignoring the issue of resonances raised in the point above).

---

## Round 1 · Referee Report · Anonymous (Referee 2) · 2019-11-18

Report

The authors discuss a hierarchy of topological bulk invariants in anomalous Floquet insulators, which they introduced in Ref. 38. The anomalous Floquet insulator is a two dimensional, periodically driven MBL system, and assuming a phenomenology identical to 1D MBL systems based on local integrals of motion (LIOM), the authors show that on each plaquette of the lattice, the time averaged magnetization density is a quantized, topological quantity. Since its values scale with the system size, they suggest a "natural basis" given by the expansion of the magnetization density in the basis of LIOMs. This leads to the introduction of the invariants $\mu_i$, where $\mu_i$, $i>1$ is argued to be zero in the non-interacting case, and therefore nontrivial phases in interacting systems are expected to exhibit nonzero invariants with larger $i$. An example for such a scenario is presented in a model with correlated hoppings. In the last part of the paper, a numerical simulation of the model with a five step drive (Fig. 3) is provided. For this, a set of random wavefunctions in the five particle sector is time evolved according to the driving protocol and the total magnetization $M$ (sum over all plaquette magnetizations) is easured. At long times, on average $M=5$ is found, which leads to the conclusion that $\mu_1=1$ and $\mu_i=0$ for $i>1$, due to the combinatorial expression of $M_0$ in termsof the topological invariants.

This is a very interesting paper which merits publication in SciPost Physics. I have a few comments which should be addressed prior to publication:

  1. The entire line of arguments in this paper rests on the assumption that MBL exists in 2d and that an effective description in terms of LIOMs exists. While this is a reasonable starting point, it seems at odds with recent arguments (avalanche theory) [Phys. Rev. B 95, 155129] which seem to preclude the stability of MBL in 2d. The authors comment very briefly that full MBL is not necessary, and rather partial MBL is sufficient, meaning that only some LIOMs exist (and I assume the other complementary operators become extended). I am not sure if such a description would be valid in the case of the avalanche picture, but maybe the authors can add a comment on this. Unfortunately I am not sure the numerical evidence helps here to make the point, since the observation of MBL in small system with only very few particles does not suffice to argue for the existence of MBL in 2d. This being said, the entire discussion in the manuscript is consistent in itself (including the numerics) and I don't think this comment is an obstacle for publication.

  2. In the same vein as my above comment, I am wondering about the role of the thermal edge states which the authors mention. If the edges really are thermal, would they not be a seed for a thermalization avalance? It seems to me that the edges could have a much stronger effect than the usually discussed random thermal inclusions in the system (which should be present here too). I understand that the authors leave the investigation of the edges to future work, but maybe it is possible to comment on this point. On the torus, investigated in this manuscript, this problem does not exist.

  3. The authors argue that interacting systems exhibit nontrivial topological phases different from the noninteracting case which are visible in a nonzero $\mu_i$, $i>1$ and provide an example in Sec III. Is there a reason for the choice of a different model in Sec. IV. which yields $\mu_1=1$, $\mu_i=0$ for $i>1$ in the numerical simulations? It would be nice to see numerical evidence for a case where higher invariants are nonzero.

  4. The numerical method used for the calculation of the topological invariants $\mu_i$ is indirect and it seems that a very high precision for the total magnetization is required. In fact even in the presented case, the numerical result does not allow to fix $\mu_4$ and $\mu_5$. Is there a more direct way to calculated the topological invariants?

---

## Editorial Decision

resubmitted